# Whole-genome resequencing of wild and domestic sheep identifies genes associated with morphological and agronomic traits

Xin Li ❑ et al.[#]

Understanding the genetic changes underlying phenotypic variation in sheep (*Ovis aries*) may facilitate our efforts towards further improvement. Here, we report the deep resequencing of 248 sheep including the wild ancestor (*O. orientalis*), landraces, and improved breeds. We explored the sheep variome and selection signatures. We detected genomic regions harboring genes associated with distinct morphological and agronomic traits, which may be past and potential future targets of domestication, breeding, and selection. Furthermore, we found non-synonymous mutations in a set of plausible candidate genes and significant differences in their allele frequency distributions across breeds. We identified *PDGFD* as a likely causal gene for fat deposition in the tails of sheep through transcriptome, RT-PCR, qPCR, and Western blot analyses. Our results provide insights into the demographic history of sheep and a valuable genomic resource for future genetic studies and improved genome-assisted breeding of sheep and other domestic animals.

---

[#]A list of authors and their affiliations appears at the end of the paper.

Sheep (*Ovis aries*) is an important livestock species, which has provided meat, wool, skin, and milk for humans since the Neolithic. Characterization of genome-wide sequence variation and identification of phenotype-associated functional variants are essential steps for guidance of genome-assisted breeding in the near future. The impact of domestication and subsequent selection on genomic variation has recently been investigated in sheep[1,2], and a number of quantitative trait loci (QTLs) and functional genes have been associated with phenotypic traits[3]. However, most of these investigations focused on a few phenotypes and involved a limited number of molecular markers and breeds/populations. So far, whole-genome resequencing has allowed the identification of genomic variants involved in domestication and genetic improvement for several domestic plants (e.g., rice and soybean)[4,5] and animals (e.g., cattle and sheep)[1,2,6].

The completion of a sheep reference genome[7] has allowed comparison of the genomes from a wide collection of phenotypically diverse authentic landraces and improved breeds of domestic sheep with their wild ancestors. In this study, we resequence the genomes of 16 Asiatic mouflon, 172 sheep from 36 landraces and 60 sheep from six improved breeds to a depth of ~25.7× coverage. Tests for selective sweeps and genome-wide association studies (GWAS) identify a number of selected regions and genes potentially affected by domestication and associated with several important morphological and agronomic traits. Moreover, we conduct a survey of non-silent single-nucleotide polymorphisms (SNPs) and gene-containing copy number variations (CNVs), which are part of the selective signatures. These data provide a valuable genomic resource for facilitating future molecular-guided breeding and genetic improvement of domestic sheep, potentially valuable in the face of ongoing climate change and consequent impacts in agricultural practice. In addition, our findings contribute to further understanding of the demographic history of sheep and the molecular basis of distinct phenotypes in the species and other animals.

## Results

**Sequencing and variation calling.** Deep resequencing of the 248 samples of wild and domestic sheep (Fig. 1a and Supplementary Data 1) generated a total of 137.0 billion 150-bp paired-end reads (20.55 Tb), with an average depth of 25.7× per individual and an average genome coverage of 98.27%. The average sequence coverage was 27.71× (23.90–36.93×) for 16 Asiatic mouflon, 25.23× (17.15–31.35×) for 172 landraces and 26.51× (24.62–32.98×) for 60 improved sheep (Supplementary Fig. 1 and Supplementary Data 2). There was no significant difference in sequence coverage among individuals from the three groups (Kruskal-Wallis, $P > 0.05$). Of the Asiatic mouflon and domestic sheep sequencing reads, 99.04% were mapped to the *O. aries* reference genome for both datasets. We obtained a total of 67,314,959 and 91,772,948 SNPs after mapping with SAMtools and GATK, respectively, of which 50,520,459 were identified by both methods (Supplementary Data 3 and Supplementary Notes). After filtering, a final set of 28.36 million common SNPs was retained (6.69 million/individual in domestic sheep versus 8.40 million/individual in Asiatic mouflon; Mann-Whitney, $P < 0.001$) along with 4.80 million insertions and deletions (INDELs ≤100 bp; 1.16 million/individual for domestic sheep versus 1.38 million/individual for Asiatic mouflon; Mann-Whitney, $P < 0.001$) (Supplementary Fig. 1 and Supplementary Data 3) in the downstream analyses. In addition, the high-depth whole-genome sequencing data enabled us to identify 13,551 autosomal CNVs (176 bp–224.6 kb; 311–804/individual; Supplementary Data 4) and 28,973 autosomal structural variations (SVs, 50 bp–984.0 kb;

**Table 1 Summary information of whole-genome variations identified in Asiatic mouflon, landraces, and improved breeds.**

| Variations | Asiatic mouflon | Landraces | Improved breeds |
|---|---|---|---|
| SNPs | 23,269,423 | 14,382,975 | 14,008,509 |
| Indels | 3,501,571 | 3,481,234 | 2,743,640 |
| Insertions | 1,351,550 | 1,343,664 | 1,099,340 |
| Deletions | 2,150,021 | 2,137,570 | 1,644,300 |
| SVs | 16,970 | 25,089 | 19,282 |
| CNVs | 7,331 | 12,724 | 10,694 |

4,515–6,657/individual; Supplementary Data 5) across all samples of wild and domestic sheep.

On average, 96.21% SNPs identified in the 232 domestic sheep and 81.26% SNPs identified in the 16 Asiatic mouflon were confirmed by the sheep dbSNP database v.151 (Supplementary Data 6). For the SNPs on the Ovine HD chip, an average of 98.98% genotypes identified in the sequenced samples were also validated by Ovine Infinium® HD SNP BeadChip data available for 223 individuals (Supplementary Data 7). Using 10,007 homozygous reference loci on the Ovine HD BeadChip for 211 individuals, false-positive SNP calling rates of 6.38% and 5.37% were observed for GATK and SAMtools, respectively. After filtering, the false-positive rate for the SNP set identified by both methods was estimated to be 0.66%. Moreover, inspection of 68 randomly selected SNPs in candidate genes from 1,414 individuals of 21 breeds obtained by Sanger sequencing produced an overall validation rate of 95.69% (Supplementary Table 1 and Supplementary Methods). For PCR and qPCR validation of CNVs, we confirmed 78.79% concordant genotypes (36/48 deletions and 26/33 duplications; Supplementary Fig. 2, Supplementary Data 8 and Supplementary Methods). The high-quality genomic variants generated here added ~230,000 new SNPs to the public database of genetic variants for domestic sheep.

**Patterns of variation.** The 28.36 million SNPs were analyzed across the three groups of sheep (Asiatic mouflon, landraces, and improved breeds). A majority up to 23.27 million SNPs were observed in Asiatic mouflon at 7.77–9.16 million per individual (12.06 million to be unique for this group), followed by 14.38 million in landraces at 5.62–8.92 million per individual (1.06 million to be unique) and 14.01 million in improved breeds at 5.90–6.90 million per individual (1.08 million to be unique) (Fig. 2, Table 1, Supplementary Table 2, and Supplementary Data 3). Using the Asiatic mouflon reference genome (ftp://ftp.ebi.ac.uk/pub/databases/nextgen/ovis/assembly/mouflon.Oori1.PRJEB3141/) for SNP calling, we identified 28.75 million SNPs in Asiatic mouflon, which was higher than that based on the sheep reference genome Oar v.4.0 (23.27 million).

We observed 12.09 million SNPs shared between landraces and improved breeds, which exceeded that shared between the mouflon and landraces (10.38 million) or between the mouflon and improved breeds (9.98 million) (Fig. 2 and Table 1). Pairwise genome-wide $F_{ST}$ values also indicated that genomic differentiation between landraces and improved breeds ($F_{ST} = 0.032$, $P = 0.041$) was less than that between Asiatic mouflon and landraces ($F_{ST} = 0.125$, $P = 0.015$) or between Asiatic mouflon and improved breeds ($F_{ST} = 0.132$, $P = 0.016$). The genomic diversities ($\pi$) in Asiatic mouflon, landraces, and improved breeds based on the SNPs with <10% missing data were 0.00127, 0.00113, and 0.00109, respectively. The distribution of SNPs at various regions near or within genes was similar in Asiatic mouflon, landraces, and improved breeds (Supplementary

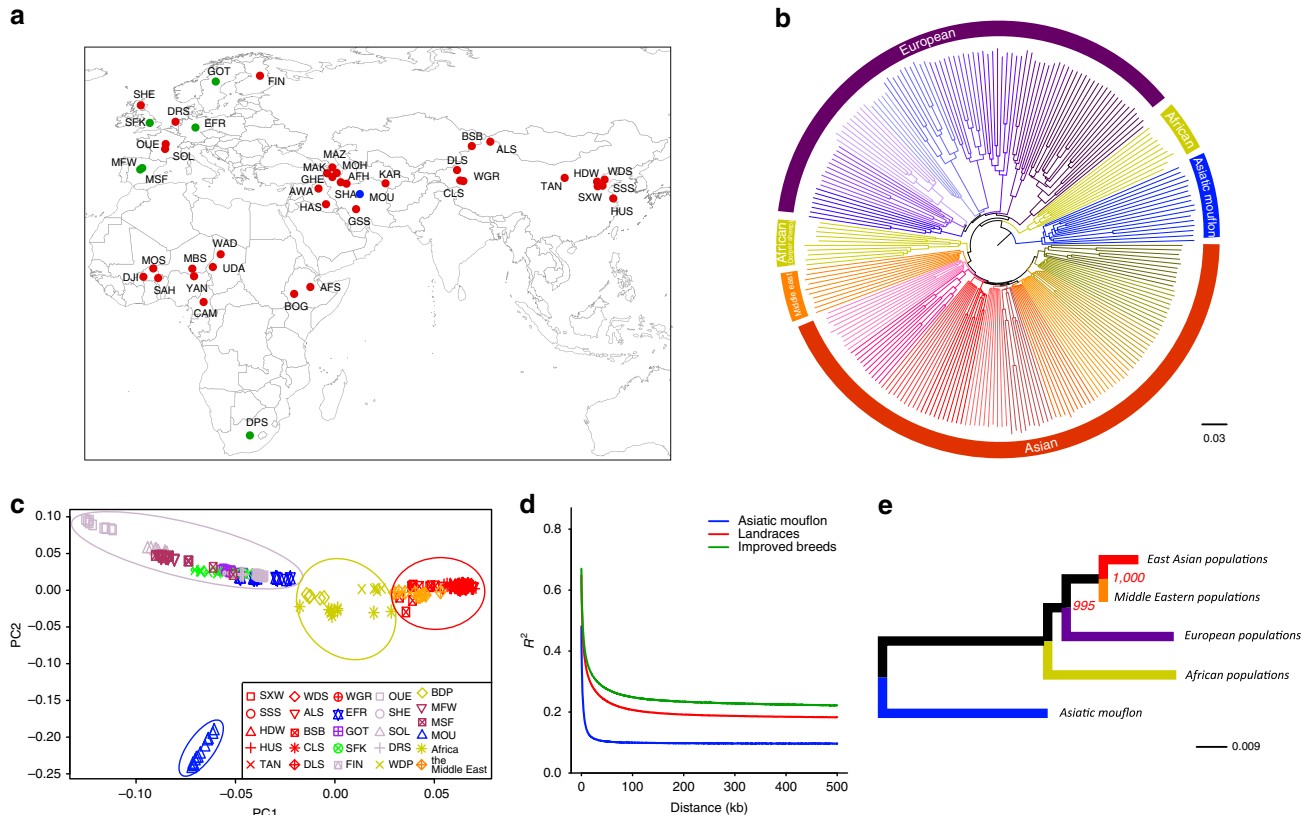

**Fig. 1 Geographic distribution and genetic structure of domestic and wild sheep. a** The geographic distribution of Asiatic mouflon, 36 landraces, and six improved breeds, which are represented by blue, red, and green dots on the world map[79], respectively. MOU, Asiatic mouflon; AFH, Afshari; AFS, Afar; ALS, Altay; AWA, Awassi; BOG, Bonga; BSB, Bashibai; CAM, Cameroon; CLS, Celle Black; DJI, Djallonké; DLS, Duolang; DPS, Dorper (WDP, white head; BDP, black head); DRS, Drenthe Heathen; EFR, East Friesian Dairy; FIN, Finnsheep; GHE, Ghezel; GOT, Gotland; GSS, Gray-Shiraz; HAS, Hamdani; HDW, Large-tailed Han; HUS, Hu; KAR, Karakul; MAK, Makui; MAZ, Mazekh; MBS, Mbororo; MFW, Chinese Merino (fine wool); MOH, Moghani; MOS, Mossi; MSF, Chinese Merino (super-fine wool); OUE, Ouessant; SAH, Sahelian; SFK, Suffolk; SHA, Shal; SHE, Shetland; SOL, Solognote; SSS, Sishui Fur; SXW, Small-tailed Han; TAN, Tan; UDA, Uda; WAD, West African Dwarf; WDS, Wadi; WGR, Waggir; and YAN, Yankasa. **b** Neighbor-joining (NJ) tree of the 248 individuals constructed using the $p$-distances between individuals, with Asiatic mouflon as an outgroup. **c** Plots of principal components 1 and 2 for the 248 individuals. **d** Decay of linkage disequilibrium in the Asiatic mouflon, landraces, and improved breeds. **e** Neighbor-joining tree of five genetic groups based on the Reynolds genetic distances. Red numbers beside divergence nodes are bootstrap values based on 1,000 replications. A scale bar represents branch length in terms of percent divergences (%). Source data are provided as a Source Data file.

Table 2). The ratio of non-synonymous and synonymous substitutions in wild (0.55) and domestic sheep (0.53–0.54) was comparable.

To ascertain the effect of major evolutionary transitions (e.g., domestication and intensive artificial breeding) on CNVs and SVs[8], these variants were pooled for Asiatic mouflon, landraces, and improved breeds separately (Fig. 2 and Table 1). This yielded a high depth of coverage and information about the uniqueness and sharing of CNVs and SVs among the three groups (Fig. 2 and Table 1). The abundance of CNVs per individual ranged from 443 to 804 (average of 563) for Asiatic mouflon, from 311 to 777 (average of 589) for landraces, and from 514 to 686 (average of 616) for improved breeds (Supplementary Data 4). The number of SVs per individual varied between 5,035 and 6,657 (mean = 5,874) in Asiatic mouflon, between 4,515 and 6,323 (mean = 5,393) in landraces, and between 4,863 and 6,203 (mean = 5,366) in improved breeds (Supplementary Data 5). In contrast to the abundance of SNPs identified in the wild species, we detected significant differences in the numbers of CNVs (Kruskal-Wallis, $P = 0.047$) and SVs (Kruskal-Wallis, $P < 0.001$) per individual among the three groups (Supplementary Data 4 and 5).

From a total of 6,929 common CNV regions (read-depth signal value <0.3 or >1.7 for all 248 individuals; see Online Methods), we

found 946 functional genes overlapping with 1,999 CNV regions (Supplementary Data 9 and Supplementary Notes). The top 15 significant Gene Ontology (GO) terms and Kyoto Encyclopedia of Genes and Genomes (KEGG) pathways for the 6,220 unique CNV regions in the 232 domestic sheep were enriched for biological processes involved in binding of sperm to zona pellucida and cell–cell recognition as well as for pathways associated with neural system function and immune system response (Supplementary Data 10). GO and KEGG pathway analyses for the 402 unique CNV regions in the 16 Asiatic mouflon uncovered enriched GO terms associated with adhesion that play essential roles in cell shape, motility, and proliferation as well as pathways involved in metabolism, neural system, and focal adhesion (Supplementary Data 10).

**Population structure, linkage disequilibrium, and demography**. To understand the population structure and demographic history of Old World domestic sheep, we utilized the SNP dataset (Fig. 1a) in a number of contexts and analyses as follows. Using the Asiatic mouflon as an outgroup, we produced a phylogenetic tree that divided domestic populations into four subgroups of European, Middle Eastern, Asian, and two African lineages (i.e., the Dorper sheep and the 10 African landraces)

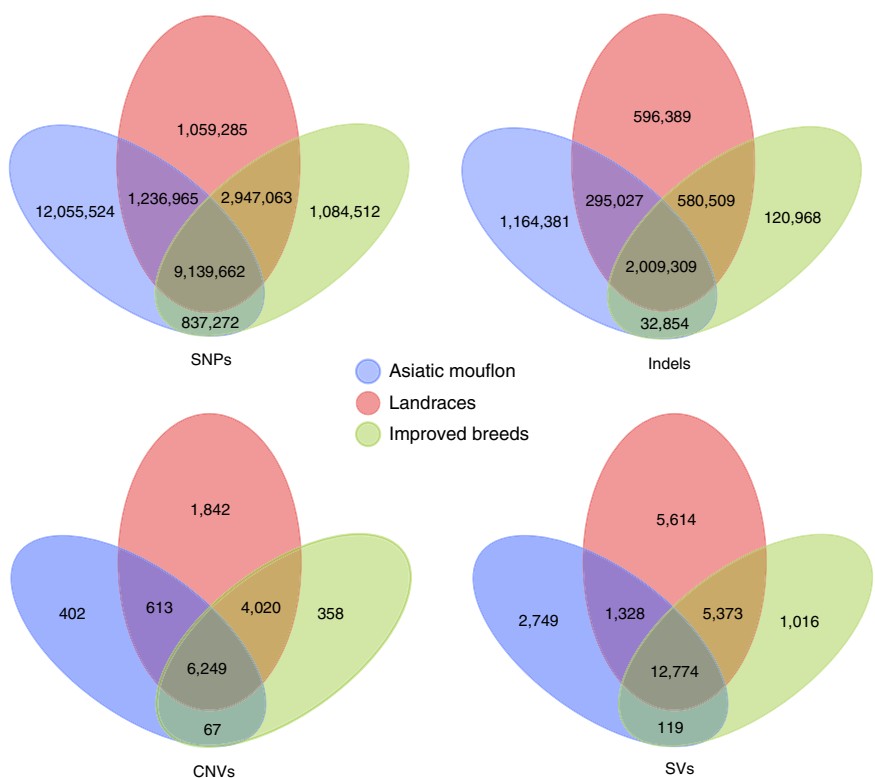

**Fig. 2 Venn diagrams summarize unique and common variants among groups.** Venn diagrams represent unique and shared SNPs, Indels, CNVs, and SVs among landraces, improved breeds, and Asiatic mouflon.

(Fig. 1b and Supplementary Fig. 3). This geographic subdivision was confirmed by principal component analysis (PCA; Fig. 1c) and clustering analysis based on maximum likelihood estimation (Supplementary Fig. 4). A neighbor-joining (NJ) tree constructed for the above four groups further revealed a close genetic affinity between East Asian and Middle Eastern populations, whereas the two African lineages showed larger genetic divergence from the other three subgroups (Fig. 1e). Nucleotide diversities ($\pi$) in European, Asian, African, and Middle Eastern populations based on the SNPs with <10% missing data were 0.00105, 0.00118, 0.00113, and 0.00114, respectively (Supplementary Fig. 5a). Asiatic mouflon were more genetically similar to Middle Eastern sheep than to other domestic populations as measured by pairwise $F_{ST}$ (Supplementary Fig. 5b).

Linkage disequilibrium (LD, measured as $r^2$) decreased to half of its maximum value at 2.8 kb in Asiatic mouflon but at 12.1 kb and 17.1 kb in landraces and improved breeds, respectively (Fig. 1d, Supplementary Tables 3 and 4 and Supplementary Notes). LD comparison among domestic populations (Supplementary Fig. 6c and Supplementary Table 3) showed that European populations had a higher level of LD (25.1 kb) than that in Asian populations (9.8 kb). Pairwise sequentially Markovian coalescent (PSMC) analysis revealed concordant demographic trajectories for wild and domestic sheep, with two expansions and two contractions in $N_e$ during the last one million years (Supplementary Fig. 7). The estimated $N_e$ for Asiatic mouflon and domestic populations ~50 generations ago[9] were 344.1 and 73.7–199.3, respectively, being inversely correlated with the extent of LD (Supplementary Fig. 5c) as expected. These observations suggested that artificial selection and genetic isolation, leading to the formation of breeds, had stronger effects on LD and $N_e$ than on nucleotide diversity.

**Genomic signatures related to domestication.** To identify genomic regions influenced by domestication, we compared the

genomes of 16 Asiatic mouflon and five old landrace populations representing different geographic and genetic origins: 5 Dutch Drenthe Heathen[10], 10 East-Asian Hu[11], 10 Central-Asian Altay[11], one African Djallonké[12], and one Middle Eastern Karakul sheep[13]. Using the cross-population composite likelihood ratio (XP-CLR) test, we scanned for genomic regions with extreme allele frequency differentiation. The top 1% XP-CLR values identified 302 putative selective sweeps in the five old landraces after annotation and removing repeats (Fig. 3a and Supplementary Data 11). As genomic regions targeted by artificial selection may be expected to have decreased levels of genetic variation, we also measured and plotted nucleotide diversity ($\pi$) along their genomes. Selecting the windows with the top 1% diversity ratios, i.e., low diversity in the five old landraces but high in the Asiatic mouflon, we found 529 putative selective sweeps (Supplementary Data 12). Combination of the XP-CLR and $\pi$ ratio analyses unveiled 144 putative selective regions covering or being near to 261 genes in the five old landraces (Supplementary Data 13). Additional analyses involving the integrated haplotype score (iHS) analysis (top 5% outliers) and the Hudson-Kreitman-Aguadé (HKA) test ($\chi^2 = 5.99$, $df = 2$, $P = 0.05$) identified 899 and 1,503 putative selective sweeps, respectively (Supplementary Data 14 and 15). Sixty-five and 71 selected genes identified by both XP-CLR and $\pi$ ratio analyses were also detected by the iHS and HKA analyses, respectively (Supplementary Data 16 and 17). A comparison of the domestication-associated selective sweeps and known QTLs[14] (permutation test, $P < 0.001$; Supplementary Table 5) revealed that the selected regions with high XP-CLR values but reduced diversity and significant values in the iHS or HKA analysis mostly spanned milk- and meat-related QTLs (Supplementary Data 18 and 19), reflecting human demands for milk and meat during sheep domestication.

Among the 261 candidate genes revealed by two (XP-CLR and $\pi$ ratio) or three (XP-CLR, $\pi$ ratio, and iHS or HKA) methods, 36

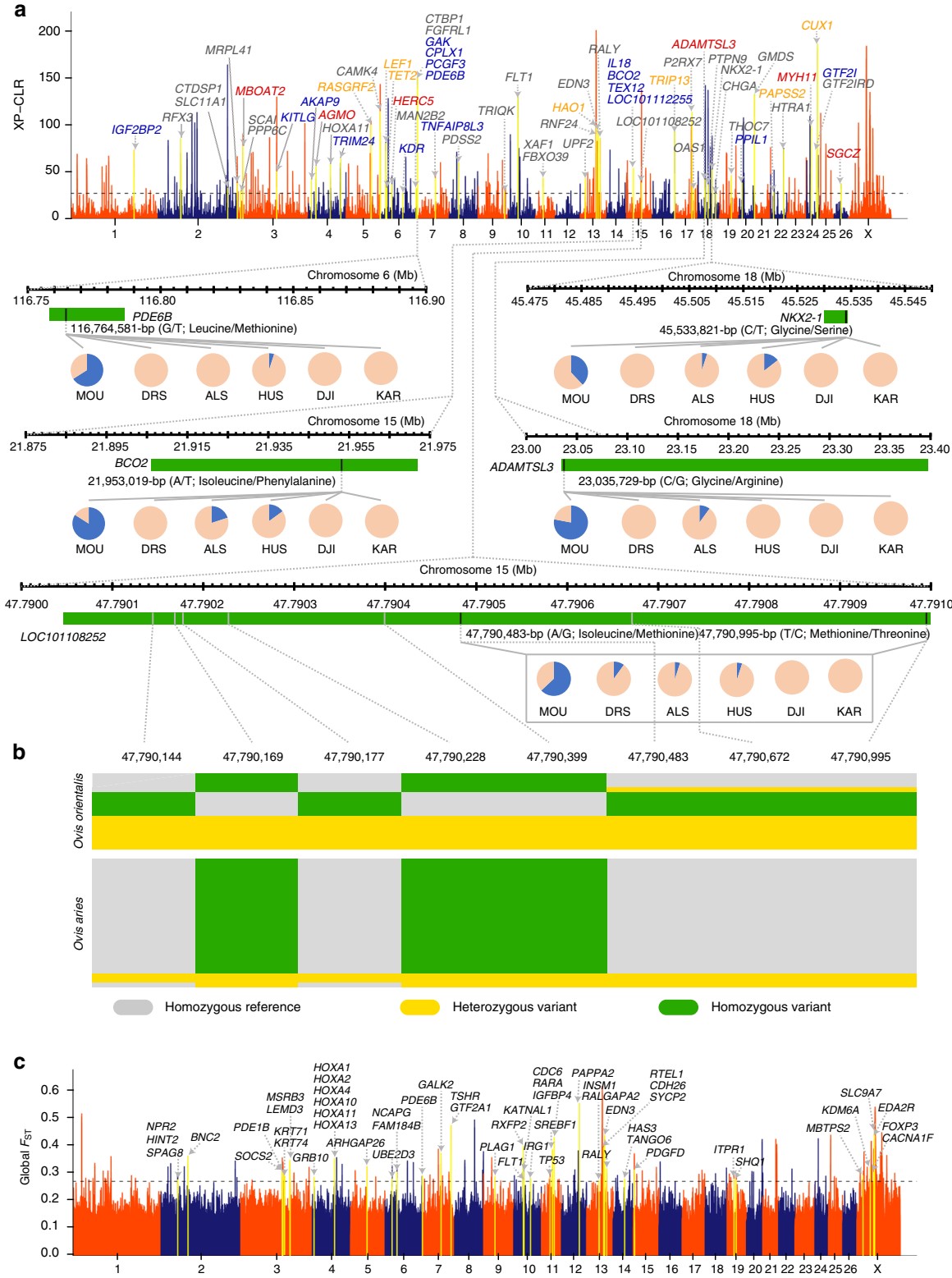

were also identified to be the targets of selection in the comparison of Asiatic mouflon with domestic sheep in two recent studies (Supplementary Table 6)[1,2]. In the same selection tests (XP-CLR and π ratio) between the Asiatic mouflon and the five old landraces, none of the 48 selected genes in the Asiatic mouflon (Supplementary Data 20) was found in the 261 selected genes in the five old landraces. Diverged selection has thus driven the domestic sheep away from the Asiatic mouflon,

and the 36 consistently selected genes identified in domestic sheep were plausibly linked to domestication (Supplementary Table 6).

Inspection of the 261 selected genes in the five old landraces detected 14 (*SLC11A1*, *HOXA11*, *CAMK4*, *LEF1*, *TET2*, *KDR*, *CTBP1*, *GAK*, *CPLX1*, *PCGF3*, *FLT1*, *BCO2*, *CHGA,* and *HTRA1*) to be associated with known functions (e.g., female reproductive traits, resistance to infection, bone formation, fat deposition,

**Fig. 3 Genome-wide annotations during sheep domestication and improvement. a** Whole-genome screening for selected regions during domestication by comparing five old landrace populations (Drenthe Heathen (DRS) in Europe, Altay (ALS) in Central Asia, Hu sheep (HUS) in East Asia, Djallonké sheep (DJI) in Africa, and Karakul sheep (KAR) in the Middle East) with Asiatic mouflon (MOU) through the XP-CLR. The black horizontal dashed line corresponds to the genome-wide significance threshold (XP-CLR = 26.96). Candidate genes overlapping with regions which were significantly selected by XP-CLR & ln($\pi$ ratio)/ln(2), XP-CLR & ln($\pi$ ratio)/ln(2) & iHS, XP-CLR & ln($\pi$ ratio)/ln(2) & HKA, and XP-CLR & ln($\pi$ ratio)/ln(2) & iHS & HKA are marked by gray, orange, blue, and red colors, respectively. Below this plot genes near the peaks are indicated by green boxes. The pie charts represent the spectrum of allele frequencies at the non-synonymous loci of the focused genes *PDE6B*, *BCO2*, *NKX2-1*, *ADAMTSL3*, and *LOC101108252* in Asiatic mouflon and the five old landraces. The type of variant allele is indicated in blue, whereas the reference allele in pink. **b** The patterns of genotypes of the *LOC101108252* gene region among Asiatic mouflon and the five old landraces based on eight SNPs. **c** Genome-wide distribution of global $F_{ST}$, which is measured by the average value for each SNP across all 42 domestic breeds. The significance threshold ($F_{ST} = 0.27$) is denoted by black dashed line. Source data are provided as a Source Data file.

yellow fat, photoperiod, and recombination rate variation) in sheep in previous studies (Supplementary Data 21 and Supplementary Notes). Twenty-two other genes (e.g., *IGF2BP2*, *RFX3*, *MRPL41*, *KITLG*, *HERC5*, *MAN2B2*, *FGFRL1*, *PDE6B*, *EDN3*, *RALY*, *GTF2I*, *GTF2IRD1*, etc.) were previously found to be influenced by selection in other species including cattle, goat, horse, pig, dog, chicken, rabbit, and mice (Supplementary Data 21). These genes were associated with functions including immunity, pigmentation, coat color, photoreceptor, behavior, growth, and reproduction traits (Supplementary Data 21 and Supplementary Notes). We identified non-synonymous SNP mutations in the 59 most plausible domestication genes presented above (i.e., summing the unique genes in Supplementary Table 6 and Supplementary Data 21) and found that the variant allele frequencies of non-synonymous SNPs in five genes (*PDE6B*, *BCO2*, *ADAMTSL3*, *NKX2-1*, and an olfactory receptor 51A4-like gene *LOC101108252*) and the genotype pattern in *LOC101108252* showed significant differences (Mann-Whitney, $P < 0.01$) between the Asiatic mouflon and the five old landraces (Fig. 3a, b and Supplementary Table 7).

In the functional enrichment analysis of the 261 genes putatively influenced by domestication, we identified the top 15 over-represented GO terms and 12 KEGG pathways (Supplementary Data 22). Specifically, four biological process GO terms and one KEGG pathway were associated with biosynthesis. Five biological process GO terms and four KEGG pathways were associated with metabolic processes. One KEGG pathway was associated with olfactory transduction.

In the selective sweep analysis of CNVs, we identified 137 candidate selected CNVs associated with domestication (Supplementary Data 23). Annotation of the CNVs indicated the CNVs to be located in genes (Supplementary Table 8) or coincident with known QTLs[14] (Supplementary Data 24), which are functionally related to traits and biological processes such as follicular development and fertility (*SLIT2*)[15], milk production (*JAK2*)[16], wool production (*KIF16B*)[17], adipogenesis (*TCF7L1* and *BCO2*)[18,19], and spleen size, oxygenated red blood cells and consequently high tolerance to hypoxia (*PDE10A*)[20] (Supplementary Notes). Also, we found divergent frequency distributions for seven deletions (overlapping with *RFX3*, *AGMO*, *BCO2*, *LOC101112255*, *ADAMTSL3*, and *SGCZ*) and three translocations (overlapping with *GTF2I*, *CAMK4*, and *SGCZ*) between the Asiatic mouflon and domestic sheep (Supplementary Table 9 and Supplementary Notes). Additionally, by comparing these 137 candidate domestication CNVs with the 144 domestication sweeps identified using both XP-CLR and $\pi$ ratio analyses, we detected CNVs located within two selective sweeps and annotated three genes (i.e., *BCO2*, *USP6NL*, and *LOC101112255*; Supplementary Table 10).

**Selective signatures during breeding and improvement.** After domestication, selective signatures in sheep are expected to be

engendered in different breeds through adaptation to a diverse range of environments and specialized production systems during breeding and improvement (Supplementary Fig. 8)[9,21]. In this context, we further compared the genomes of domestic breeds (i.e., the 36 landraces and six improved breeds; Supplementary Data 1) to detect signatures of positive selection during this process.

We calculated global $F_{ST}$ among the domestic breeds using a 50 kb sliding window and shift of 25 kb across genome, and identified 205 putatively selected genomic regions (Fig. 3c and Supplementary Data 25) with the top 1% global $F_{ST}$ values, which spanned 23.80 Mb and comprised 391 genes (Supplementary Data 26). Annotation of these genes revealed functions associated with phenotypic and production traits including presence or absence of horns, pigmentation, reproduction, and body size (Supplementary Table 11 and Supplementary Data 27). We also observed genes functionally related to environmental adaptation, energy metabolism, and immune response, which may have been the targets of long-term natural selection[21,22]. Functional analysis of the 391 selected genes revealed significant enrichments for GO categories involved in four biological process categories including immune response, and immune system processes as well as 11 molecular function categories, such as cytokine activity and ATPase activity, coupled to transmembrane movement of ions, phosphorylative mechanism associated with energy metabolism, and immune function (Supplementary Data 28). The most significantly enriched pathway was cytokine–cytokine receptor interaction (Supplementary Data 28).

Notably, the selective region with the highest $F_{ST}$ value ($F_{ST} = 0.56$) was located near the gene *PAPPA2*, which has been reported to be associated with fat deposition in humans[23] and has been identified as a candidate gene for milk, reproduction, and body size traits in cattle[24]. Comparison of allele frequencies at non-synonymous SNPs in candidate selected genes revealed four (e.g., *SPAG8*, *FAM184B*, *PDE6B*, and *PDGFD*) with significantly differentiated allele frequencies among domestic sheep breeds (Supplementary Data 29).

Of these 391 selected genes, nine were also among the previously identified 59 most plausible domestication genes (Supplementary Table 6 and Supplementary Data 21) and they (*RASGRF2*, *FBXO39*, *XAF1*, *GMDS*, *HOXA11*, *PDE6B*, *FLT1*, *EDN3*, and *RALY*) (Supplementary Table 11) were linked to both domestication and breed-level genetic differentiation. In addition, 22 selected genes were confirmed to be under selection in our analyses for specific phenotypic traits such as reproduction, presence of horns, fat tail, wool fineness, nipple number, and ear size (see below; Supplementary Table 11). Moreover, 52 selected genes (e.g., *SOCS2*, *EDA2R*, *PDE1B*, *PDGFD*, *HOXA10*, etc.) have been shown to be under selection in sheep, humans, and other domestic animals in previous investigations (Supplementary Data 27). Additionally, a comparison of the selected genomic regions with previously reported QTLs[14] revealed 131 regions

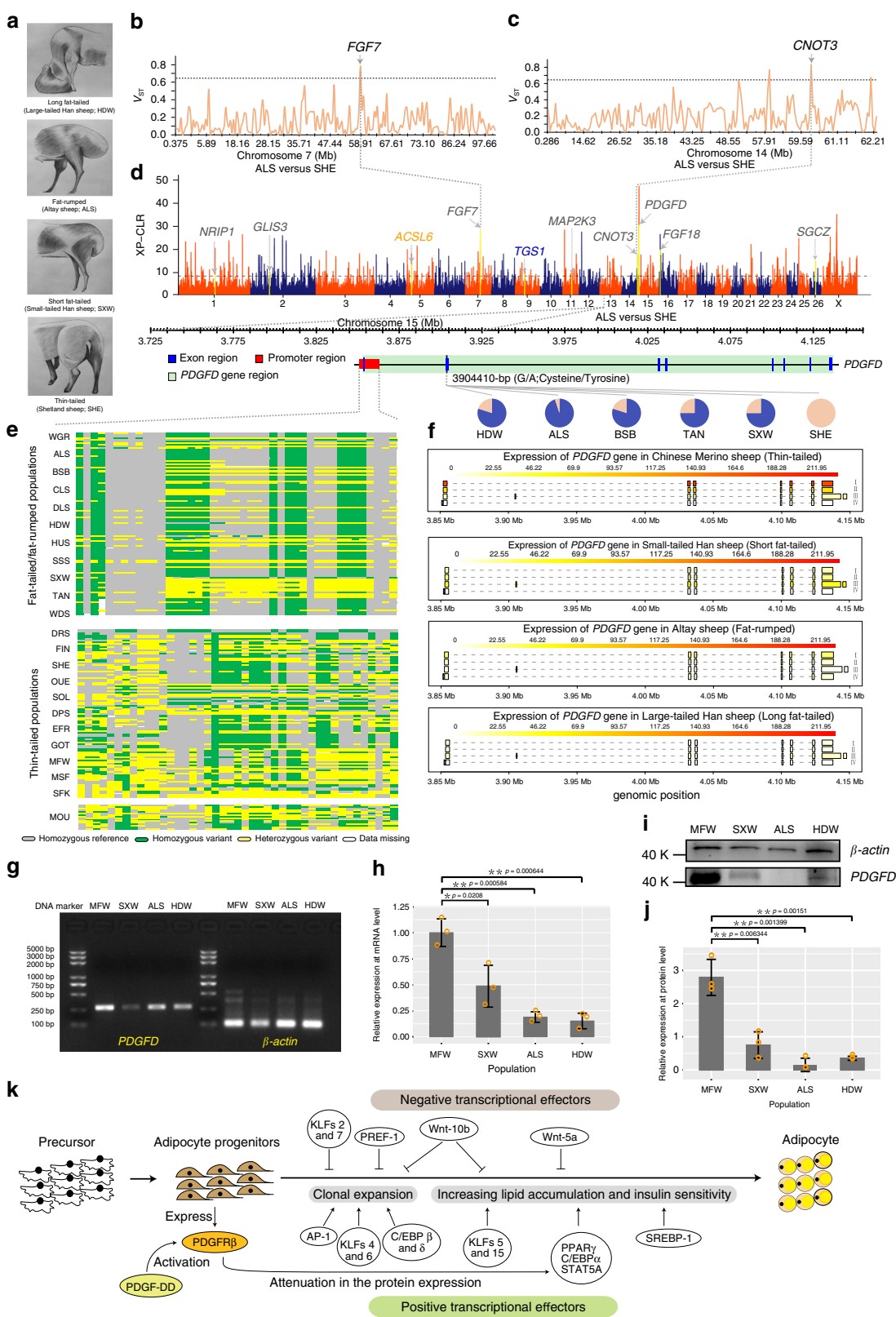

with high $F_{ST}$ values spanning the QTLs. These regions covered 19.97 Mb of the sheep genome and were found to be associated with morphological and production traits, such as reproductive seasonality, milk related traits, body weight, meat-related traits, teat number, tail fat deposition, and presence of horns (Supplementary Data 30).

**Genetic mechanisms of the tail configuration trait.** We implemented genome-wide selection tests between domestic breeds representing contrasting phenotypes for several traits that are relevant for sheep husbandry (Supplementary Table 12), such as morphological traits. Focusing on an iconic trait, tail configuration (Fig. 4a), we performed separate pairwise-population

**Fig. 4 Genome-wide screening and genetic basis of *PDGFD* for tail configuration. a** Different phenotypes in tail configurations; picture credit: Xin Li. **b, c** Statistic $V_{ST}$ is plotted for selected CNVs through pairwise comparison on chromosomes 7 **b** and 14 **c** with same threshold $V_{ST}$ value ≥ 0.64. **d** Selective regions associated with tail configuration by XP-CLR using the SNP data with the threshold XP-CLR ≥ 8.26. Candidate genes overlapping with the regions, which are significantly selected by XP-CLR & ln(π ratio)/ln(2), XP-CLR & ln(π ratio)/ln(2) & *iHS,* and XP-CLR & ln(π ratio)/ln(2) & HKA are marked by gray, orange, and blue colors, respectively. Below this plot, genes near the peaks are indicated by green boxes. The pie charts represent the spectrum of allele frequencies at the non-synonymous loci of *PDGFD* in populations of different tail configurations. The type of variant allele is indicated in blue, while the reference allele in pink. **e** Genotype patterns for the promoter region of *PDGFD* among 11 fat-tailed/rumped, 11 thin-tailed sheep, and Asiatic mouflon. **f** Structures and expression levels of four isoforms of *PDGFD*. Expression levels are shown in varying shades of yellow color. **g, i** Expression pattern of control gene *β-actin* and target gene *PDGFD* in tail fat examined by RT-PCR **g** and western blot analysis **i. h, j** The relative expressions of *PDGFD* in tail fat by real-time PCR (qPCR) **h** and western blot analysis **j. k** Adipogenesis signaling pathway[46] and the inhibitory function of *PDGFD* in differentiation of white adipocytes[45] by activating PDGFRβ signaling[44]. All experiments were repeated three times with similar results. Samples derived from the same experiment and the blots were processed in parallel. **g–j** Experiments were performed with the control sample (the thin-tailed sheep; MFW) and target samples (long fat-tailed sheep (HDW), fat-rumped sheep (ALS) and short fat-tailed sheep (SXW)). The data in **h** and **j** are presented as the mean ± SD, $n = 3$ biologically independent samples; groups with significant differences (*$P < 0.05$; **$P < 0.01$) were performed by two-tailed unpaired *t*-test. Source data are provided as a Source Data file.

selection tests through comparisons of fat-rumped (Altay (ALS) and Bashibai (BSB)), long fat-tailed (Large-tailed Han sheep (HDW)) and long wooly tailed (Drenthe Heathen sheep (DRS)) breeds with short fat-tailed (Tan sheep (TAN)) and short thin-tailed (Shetland sheep (SHE)) breeds. We selected regions with differences in allele frequencies by XP-CLR (Supplementary Data 31 and Supplementary Fig. 9) and reduced π values (Supplementary Data 32) in the pairs of breeds of ALS versus SHE, BSB versus SHE, HDW versus SHE, HDW versus TAN, and DRS versus SHE, and detected 105, 81, 88, 101, and 122 common selective sweeps that overlapped with annotated genes, respectively (Supplementary Data 33). Among these sweeps, we identified 21, 22, 18, 25, and 17 (Supplementary Data 34) and 16, 4, 5, 15, and 74 (Supplementary Data 35) sweeps overlapping with the selective signals detected by the *iHS* analysis (Supplementary Data 36) and the HKA test (Supplementary Data 37), respectively. Of these sweeps identified, we focused on genes involved in fat deposition and hair growth, and annotated functional genes with high credibility (Supplementary Data 38), including some previously reported (e.g., *PDGFD, NRIP1, KRT5,* and *KRT71*) and novel (e.g., *XYLB, TSHR, SGCZ, CNOT3, CFLAR, GLIS3, MSRA, MAP2K3,* and *FGF7*) genes.

We dissected the genomic architecture of the selected genes by calculating the frequency of the variant allele at non-synonymous SNPs. The frequencies of one variant allele each at genes *PDGFD, XYLB, TSHR,* and *SGCZ* as well as the genotype pattern located in the promoter region of *PDGFD* were different (Mann-Whitney, $P < 0.001$) between the fat-tailed (e.g., HDW, ALS, BSB, and TAN) and thin-tailed (e.g., SHE, Gotland sheep (GOT) and Finnsheep (FIN)) breeds (Fig. 4d, e, Supplementary Fig. 10 and Supplementary Data 39).

Remarkably, *PDGFD* was consistently selected by multiple comparisons (Supplementary Data 38). Transcriptome analysis among populations with different tail configurations (Supplementary Table 13) also identified *PDGFD* as significantly differentially expressed gene (log$_2$(fold change) = 3.08; $P_{adj}$ = 0.045) between the fat-tailed/fat-rumped and the thin-tailed sheep (Supplementary Data 40). Furthermore, we detected four transcripts (i.e., transcripts I, II, III, and IV) of the *PDGFD* gene with the transcript I to be the most differentially expressed isoform between the thin-tailed and the fat-tailed/fat-rumped sheep (Fig. 4f), indicating its primary role in regulating fat deposition in tail. Furthermore, RT-PCR, qPCR, and western blot analyses demonstrated that gene expression level and protein level of *PDGFD* were consistently correlated negatively with fat deposition in sheep tail, with the highest level observed in the thin-tailed Merino sheep (MFW), followed sequentially by the small-tailed Han sheep (SXW), the large-tailed Han sheep

(HDW), and fat-rumped Altay sheep (ALS) (Fig. 4g–j and Supplementary Fig. 11).

**Selective and association signatures for other traits**. Apart from tail configuration, we found many selected regions, novel functional genes, and non-synonymous SNPs related to the potentially selected genes, which may be responsible for traits such as reproduction, milk yield, wool fineness, meat production, and growth rate as well as for morphological traits including numbers of horns and nipples, pigmentation, and ear size (Fig. 5, Supplementary Figs. 10 and 12–20, Supplementary Data 38 and 39 and Supplementary Notes). Also, we presented a selective sweep analysis of CNVs for nine phenotypic traits (34 pairwise comparisons between domestic breeds; Supplementary Table 12) using $V_{ST}$[25], and identified a set of trait-associated CNVs and their associated functional genes as part of the selective signatures, which are known to be responsible for the phenotypic traits (Fig. 4b, c, Supplementary Figs. 9 and 13–20, Supplementary Data 41 and 42 and Supplementary Notes). For both SNPs and CNVs, we observed quite a number of the selective sweeps overlapped with known QTLs[14] associated with several production traits (Supplementary Data 43–45 and Supplementary Notes). On top of the detection of previously known QTLs, our results also revealed several novel selective sweeps, CNVs, and genes to be potentially responsible for the trait of ear size (Supplementary Fig. 20 and Supplementary Data 38 and 42).

To fine-map regions identified using selective sweep methodologies and search for direct evidence of genotype-phenotype associations, we performed GWAS for three quantitative traits (i.e., litter size and numbers of horns and nipples) with informative phenotypic records (Supplementary Fig. 21). Using a panel of samples from multiple breeds and high-quality SNPs as well as the mixed linear model (MLM), we identified 600, 989, and 1969 significant GWAS signals for litter size (109 samples from 11 breeds; 14,574,050 SNPs), number of horns (146 samples from 15 breeds; 14,556,831 SNPs), and number of nipples (123 samples from 13 breeds; 14,415,949 SNPs) with the thresholds of $-\log_{10}(P \text{ value}) = 6, 6, 4$, respectively (Fig. 5c, Supplementary Fig. 22 and Supplementary Data 46). Furthermore, we detected 20, 56, and one of these respective GWAS signals to be overlapped with selective sweeps detected for the three traits (Supplementary Data 47), suggesting the significance of these genomic regions in shaping the traits.

Except for previously reported major candidate genes (e.g., *BMPR1B, INHBB,* and *ESR1*)[26], annotation of the significant GWAS signals revealed that those for litter size were mapped to a number of novel genes, such as *NOX4, IRF2, PDE11A, ZFAT, ZFP91, TENM1, BICC1, LRRTM3, CTNN3, SMYD3, KCNN3,* and

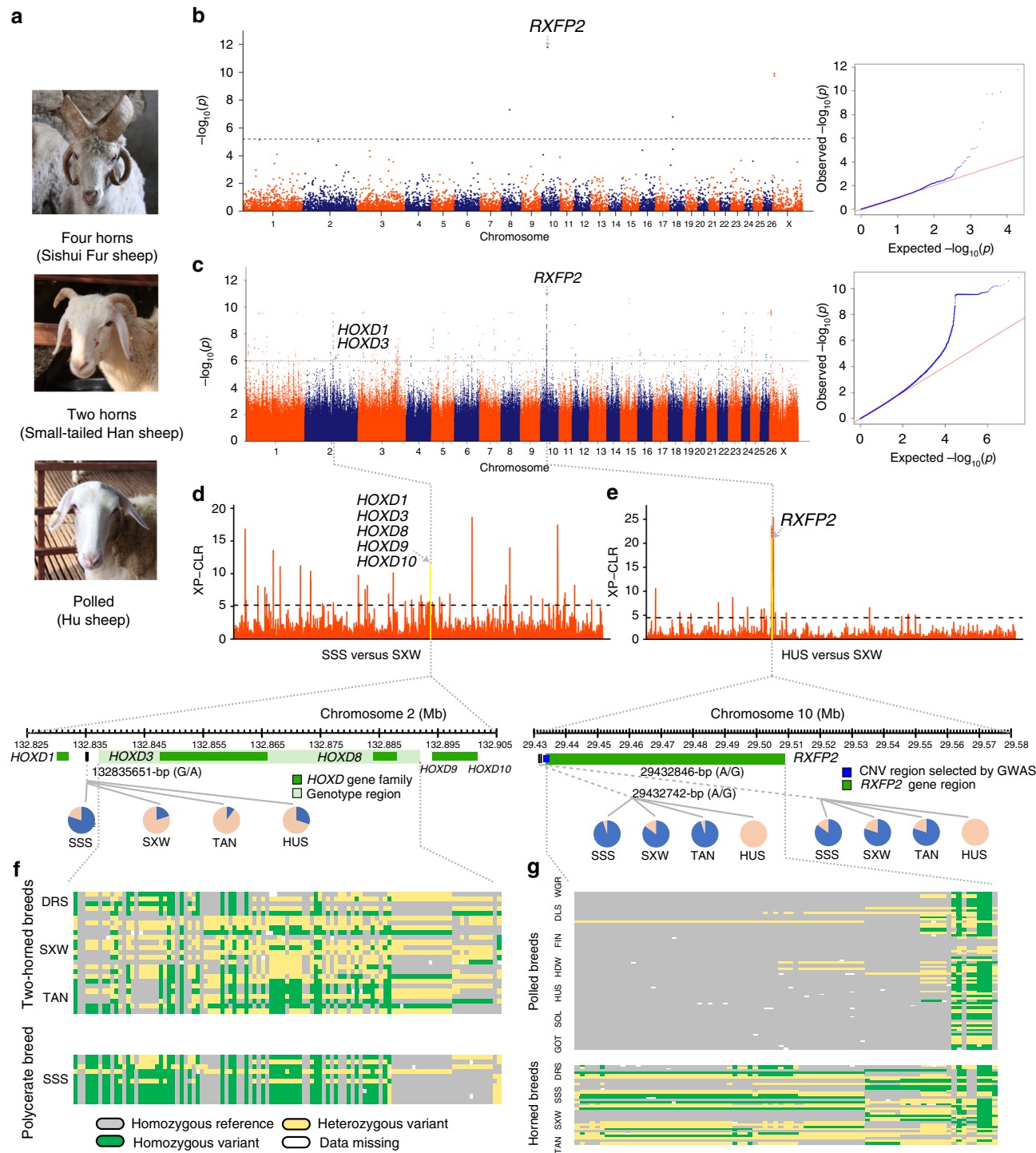

**Fig. 5 Selective and association signatures for sheep horn number. a** Different phenotypes of horn types between Sishui Fur sheep (SSS), Small-tailed Han sheep (SXW), and Hu sheep (HUS); picture credit: Meng-Hua Li. **b**, **c** Manhattan plot and quantile–quantile plot of association signals for the number of horns based on whole-genome CNV data **b** and SNPs **c**. The horizontal dashed lines correspond to the genome-wide significance thresholds ($-\log_{10}(0.05/\text{Total CNVs}) = 5.29$ for CNVs and $-\log_{10}(P\text{ value}) = 6$ for SNPs). **d** Manhattan plot of selective sweeps for polycerate trait (SSS versus SXW) on chromosome 2. Allele frequency distribution of one non-synonymous SNP at the downstream of *HOXD1* gene in one polycerate breed (SSS), two two-horned breeds (SXW and TAN) and one polled breed (HUS). The horizontal dashed line corresponds to the genome-wide significance threshold (XP-CLR = 5.17). **e** Manhattan plot of selective sweeps for polled trait (HUS versus SXW) on chromosome 10. Allele frequency distribution of one non-synonymous SNP at the downstream of *RXFP2* gene in one polycerate breed (SSS), two two-horned breeds (SXW and TAN) and one polled breed (HUS). The horizontal dashed line corresponds to the genome-wide significance threshold (XP-CLR = 4.49). In all pie chart figures, the variant allele is indicated in blue, whereas the reference allele is indicated in pink. **f** Genotype patterns of the genes *HOXD3* and *HOXD8* among one polycerate breed and three two-horned breeds. **g** Genotype patterns of the gene *RXFP2* among four horned breeds and seven polled breeds. Source data are provided as a Source Data file.

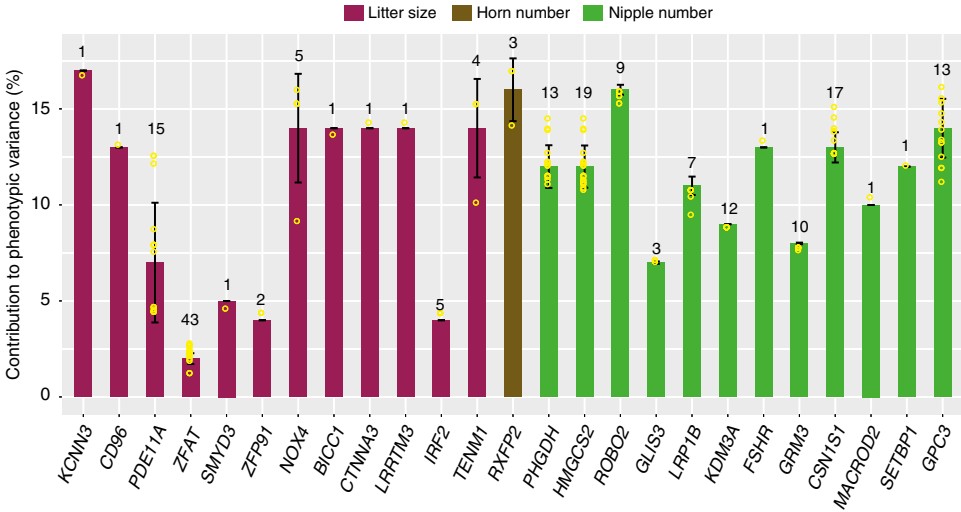

**Fig. 6 Average contributions of loci to phenotypic variances of three traits.** The identified genetic loci are located within ±20 kb of 25 candidate selected genes of the traits of litter size and numbers of horns and nipples. The number of significant loci within each gene is indicated at the top of each bar and the data are presented as the mean ± SD. The proportion of the phenotypic variation explained for each significant locus is listed in Supplementary Data 48.

*CD96* (Supplementary Fig. 22a), which play roles in various reproductive functions, including embryogenesis, uterine remodeling, follicular development and ovulation[27,28]. Focusing on 600 significant SNPs, we found 323 SNPs in intergenic regions, one in the downstream and 276 SNPs in the coding regions (only one SNP in the exon but 275 in the introns). For the number of horns, several GWAS peaks were situated in *HOXD1, HOXD3,* and *RXFP2* (Fig. 5c) as well-characterized functional genes for the polycerate and polled phenotypes in sheep[29]. For the number of nipples, most of the significant GWAS signals were located in genes associated with breast cancer, including five genes (*LRP1B, GRM3, MACROD2, SETBP1,* and *GPC3*) reported previously[30]. In particular, we detected seven novel genes (*PHGDH, KDM3A, GLIS3, FSHR, CSN2, CSN1S1,* and *ROBO2*; Supplementary Fig. 22b), which were reported to be associated with mammary and nipple development in mice[31,32].

To investigate the genetic architecture of litter size, numbers of horns and nipples, we calculated the proportion of phenotypic variation explained by the genetic variants identified in GWAS. Focusing on 189 signals located within 20 kb of 25 genes (i.e., *NOX4, IRF2, PDE11A, ZFAT, ZFP91, TENM1, BICC1, LRRTM3, CTNN3, SMYD3, KCNN3,* and *CD96* for litter size; *RXFP2* for number of horns; *LRP1B, GRM3, MACROD2, SETBP1, GPC3, PHGDH, KDM3A, GLIS3, FSHR, CSN2, CSN1S1,* and *ROBO2* for number of nipples) identified in this study, we detected 80 SNPs to explain 1.2–16.8% phenotypic variation in litter size, 106 SNPs to explain 7.0–16.1% variation in number of nipples, and three SNPs to explain 14.1–17.0% variation in number of horns (Fig. 6 and Supplementary Data 48).

Finally, we examined a total of 3,558 significant association signals with previously reported QTLs for reproduction, numbers of horns and nipples, and mapped 121 signals in five QTLs responsible for reproductive seasonality and total lambs born (permutation test, *P* < 0.001), 87 signals in one QTL associated with horns (permutation test, *P* < 0.001), and 31 signals in four QTLs related to teat placement, udder depth, udder shape and udder attachment (Supplementary Table 5 and Supplementary Data 49). We also implemented GWAS analyses using the CNV data for litter size, numbers of horns and nipples (Fig. 5b, Supplementary Fig. 23 and Supplementary Tables 14 and 15). We detected a total of 11 significant association signals and associated functional genes for the three traits (litter size: *SMARCA1* and

*APP*; number of horns: *RXFP2*; and number of nipples: *GPC5*) (Supplementary Notes). Several significant CNVs found by GWAS were in common with CNVs identified by selective sweep analysis. Two of these common CNVs were associated with reproductive traits while three CNVs with horn related traits (Supplementary Table 16).

## Discussion

In this study, we re-sequenced the whole genomes of 248 wild, landrace, and improved sheep with an emphasis on local breeds with distinct phenotypes that have not been studied previously at the genomic level. Our exploration of the sheep variome and selective sweeps focused specifically on domestication and selective breed formation.

Deciphering the genetic basis of animal domestication is an active research area. The availability of whole-genome sequences provides an opportunity to study this at the gene mutation level. Such genomic studies have been implemented in other domestic animals[33,34] and recently also in sheep[1,2]. These studies were based on genome sequences of low-to-medium coverages (8.4–17.2× in ref. [2]; 12–14× in ref. [1]). We have complemented this work using a hierarchically structured breed panel and high-depth whole-genome sequencing (mean depth of 25.7×).

We observed a lower level of genomic diversity in domestic breeds than their wild ancestors. This suggested that a substantial proportion of the genomic variation has been lost during and after domestication, whereas the genomic diversity in landraces has been largely retained in improved breeds. It should be noted that the nearly identical nucleotide diversity between landraces and improved breeds was also consistent with the observation that very strong positive selective pressure on modern breeds has only been in practice over the last ~200 generations[9]. Similar patterns of changes in genomic diversity through domestication have been observed in yak[35] and soybean[5]. The estimates of the ratio of non-synonymous and synonymous substitutions in wild (0.55) and domestic sheep (0.53–0.54) were lower than those in a previous study with a lower sequencing coverage (domestic sheep 0.66, European mouflon 0.69)[22]. The low values in both wild and domestic sheep suggested a strong impact of purifying selection.

By contrasting Asiatic mouflon to the most primitive sheep landraces in our panel, we focused on the early stage of

domestication. Results of functional enrichment analysis of the 261 candidate genes were in agreement with previous evidence that metabolic process and olfactory transduction were involved as primary functional categories in sheep domestication[2]. Furthermore, functions of the domestication-related genes and significant differences in non-synonymous SNP allele frequencies between Asiatic mouflon and domestic sheep provided additional evidence for signatures of selection on these genes and associated traits (e.g., reproduction, immunity, fat, photoreceptor, and olfaction) brought about by domestication. Nevertheless, by annotation of the 260 fixed SNPs (derived allele frequency ≥ 0.95) in the 209 putative selective sweeps, we detected 206 SNPs located in the non-coding regions but only two SNPs in the exon regions including a single non-synonymous mutation. This finding suggested domestication as a quantitative trait (e.g., litter size) to be affected mostly by mutations in non-coding regions[36,37], whereas very few mutations were non-synonymous. In addition to the functional non-synonymous SNPs, we observed differentiated frequencies in SVs between Asiatic mouflon and domestic sheep (Supplementary Tables 9). Thus, SVs in functional genes could also account for the changes in phenotypic traits during domestication. In particular, our results indicated that not only SNPs but also CNVs associated with adipogenesis (*BCO2*)[19] and proteostasis (*LOC101112255*)[38] have been under selection during domestication. Interestingly, as SVs in *GTF2I* have been found to be linked to Williams-Beuren syndrome in humans[39] and hypersocial behavior (i.e., a feature of the domestication syndrome[40]) in domestic dogs[41], the specific SV present in *GTF2I* may account for some behavioral differences between Asiatic mouflon and domestic sheep gained or lost during domestication.

This work may inform ongoing and future analysis of ancient DNA in order to pinpoint more accurately the origin and time of sheep domestication and associated impact at the genomic level. We noted that Asiatic mouflon have several subspecies (e.g., *Ovis orientalis gmelini*, *O. o. ophion*, and *O. o. laristanica*)[42] and are distributed in various geographic areas such as Iran, Turkey, Azerbaijan, and Cyprus, therefore a comprehensive sampling of all of them would be necessary in future investigations. It would also be interesting to compare genetic mechanisms responsible for specific domestication traits, and identify general patterns across different species of domestic animals[1,6].

In addition, we detected a few selective signatures associated with phenotypic and production traits during breeding and improvement. Functions of the candidate genes implied potential roles of human-induced changes in growth rate (*SPAG8*), reproduction (*FAM184B*), photoreceptor development (*PDE6B*), and tail configuration (*PDGFD*) during the development of specific breeds (Supplementary Notes). In particular, a strong selective signature located near the *PAPPA2* gene implied intense artificial selections for body fat or types of tail (e.g., fat and thin tails) and production traits in sheep towards unique breeds. It is worth noting that the most significantly enriched pathway for the 391 selected genes identified from $F_{ST}$ analysis (Supplementary Data 26) was cytokine–cytokine receptor interaction (*ACVR2B*, *TNFSF4*, *CCL25*, etc.) (Supplementary Data 28). Previous studies have revealed that, instead of artificial selection, this important pathway for immune function can also be impacted by long-term natural selection[21,22].

We investigated various phenotypic traits using an integrated analysis of whole-genome selection scans and GWAS. Although some of the candidate genes have been functionally associated with various traits in previous investigations[3], we used high-coverage sequencing data to identify a number of novel candidate genes. In particular, we mapped novel candidate genes associated with popular traits that were studied previously (e.g., litter size, ear size, and coat color) or unique traits that were rarely studied

in sheep and other livestock (tail configuration and numbers of horns and nipples). Moreover, our findings will help to narrow down the functional sub-regions within the QTLs and pinpoint the causal genes associated with these breeding-related traits. In particular, we identified several previously reported and a few novel genes to be associated with the tail configuration in sheep (Supplementary Data 38). Previous studies revealed that amino-acid changes in *PDGFD* (cysteine/tyrosine) have an important role in fat metabolism and adipogenesis in humans[43]. We envisage that non-synonymous variants with deviating allele frequencies between wild and domestic sheep and large effects on specific phenotypes in domestic sheep might be useful, for instance, as targets in CRIPSR/Cas experiments. Transcriptome analysis showed significant differential expressions of *PDGFD* transcripts among populations with different tail configurations (Supplementary Data 40). This may be ascribed to the distinct genotype pattern in the promoter region of *PDGFD* (Fig. 4e). *PDGFD* functions by causing dimerization and further activating PDGF receptor PDGFRβ[44]. Early studies showed that PDGFRβ signaling has an essential role in inhibiting differentiation of white adipocytes by regulating the expression of *PPARγ2* and *C/EBPα*[45], which were identified as the key transcriptional regulators of adipogenesis[46] (Fig. 4k). Therefore, our results provide in-depth insights into the genomic architecture and molecular mechanism for tail configuration in sheep at the genotype, variant allele, transcript, and protein levels.

All our efforts have resulted in a unique data resource in terms of the sheep variome and selective sweeps with different categories of genetic markers, allele distributions in different breeds, and associations with phenotypes with different degrees of experimental validation. This will underpin more-accurate identification of causative gene variants in the near future and facilitate novel breeding strategies, like marker-assisted or genomic selection and genome editing targeting favorable traits towards a cost-effective and environmentally friendly sheep industry.

## Methods

**Sample collection, DNA extraction, and sequencing**. Blood samples were collected from a total of 248 individuals comprising 232 domestic sheep (*O. aries*) and 16 wild sheep (Asiatic mouflon *O. orientalis*). The domestic sheep samples represent 36 landraces (172 individuals) and six improved breeds (60 individuals) with different geographic origins from Asia, Europe, Africa, and the Middle East (Fig. 1a). More specifically, the domestic samples represent various geographic origins, morphological characteristics, and production traits (Supplementary Data 1). Breed origins of the domestic sheep samples included populations from geographic areas underrepresented in earlier work (China, Afghanistan, Iran, Iraq, Azerbaijan, South Africa, Ethiopia, Burkina Faso, Niger, Nigeria, Chad, and Cameroon) as well as Germany, Spain, England, Finland, France, Scotland, Sweden, and the Netherlands (Fig. 1a and Supplementary Data 1). All the domestic sheep were typical of the breeds and unrelated according to pedigree records or herdsman's information (Fig. 1a and Supplementary Data 1). The Asiatic mouflon were collected from captivity in Iran, which is within the putative geographic center of sheep domestication. To minimize potential bias as a result of overrepresentation of local effects (e.g., inbreeding), individuals from different locations were sampled. A full description of the samples is detailed in Supplementary Data 1. Genomic DNA was extracted following the standard phenol-chloroform extraction procedure. For genome sequencing, at least 0.5 μg of genomic DNA from each sample was used to construct a library with an insert size of ~ 350 bp. Paired-end sequencing libraries were constructed according to the manufacturer's instructions (Illumina Inc., San Diego, CA, USA) and sequenced on the Illumina HiSeq X Ten Sequencer (Illumina Inc.).

**Sequence read mapping**. We obtained ~82.86 Gb of raw sequences for each sample, giving an average depth of 25.7× coverage for clean reads (17.2–37.0×) (Supplementary Data 2). The 150-bp paired-end reads were mapped onto the sheep reference genome Oar v.4.0 (https://www.ncbi.nlm.nih.gov/assembly/GCF_000298735.2) with the Burrows-Wheeler Aligner v.0.7.8 (ref. [47]) using the default parameters. Mapping results were then converted into the BAM format and sorted with SAMtools v.1.3.1 (ref. [48]). Duplicate reads were removed using SAMtools. If multiple read pairs had identical external coordinates, only the pair with the highest mapping quality was retained.

**SNP calling, validation, and annotation**. After mapping, we performed SNP calling separately for the two sets of samples (see below) using the Bayesian approach implemented in SAMtools and Genome Analysis Toolkit (GATK) v.3.7 (ref. [49]), with all individuals in each set simultaneously. One set included the 228 wild/domestic sheep with at least five samples per breed/population, which was used in all analyses, whereas the other set consisted of the 20 domestic sheep from the Middle East and Africa with one individual per breed, which was used to explore population structure and demographic history of domestic sheep in the Old World and to identify selective signatures associated with domestication and improvement (e.g., global $F_{ST}$ analysis). Only SNPs detected by both methods were kept for further analyses. The detailed processes were as follows: (i) For the GATK, the UnifiedGenotyper parameters -stand_emit_conf and -stand_call_conf were both set as 30. The same aligned BAM files were used in SNP calling through the SAMtools mpileup package; and (ii) For filtering using the command parameters –mis 0.1 –maf 0.05 –qd 2 –fs 60 –mq 40 –dp_min 6 –dp_max 120 –DP_min 10 –DP_max 30000 –gq 20 –MQRankSum -12.5 –ReadPosRankSum -8.0, the common sites were first identified by the GATK and SAMtools using the SelectVariants package, and then SNPs with missing rates ≥0.1 and minor allele frequencies (MAF) <0.05 from the three groups (Asiatic mouflon, landraces, and improved breeds) were filtered out from further analysis. For Asiatic mouflon and each breed of domestic sheep, we estimated the site frequency spectrum (Supplementary Fig. 24) based on individual genotype likelihoods assuming Hardy-Weinberg equilibrium using the ANGSD v.0.915 (ref. [50]) with the parameters –dosaf 1 –fold 1 –maxIter 100.

To validate the SNPs detected, we first compared the identified set with the *O. aries* dbSNP v.151 at the National Center for Biotechnology Information (NCBI; http://www.ncbi.nlm.nih.gov/SNP/). Next, we compared the genotypes of the called SNPs with those on the Ovine Infinium HD BeadChip array (~600 K SNPs) (Illumina, San Diego, CA) for 223 samples with available chip data (Supplementary Methods). In addition, to assess the performances of the GATK and SAMtools variant calling methods, we employed a false-positive measure by determining the rate of the monomorphic reference loci on the Ovine Infinium HD BeadChip array that were erroneously called as variant loci by the variant calling methods. We calculated the false-positive rates as the number of false heterozygous SNPs divided by the total number of homozygous reference loci[51].

SNPs were annotated using the ANNOVAR v.2013-06-21 (ref. [52]) based on the sheep reference genome Oar v.4.0 and then categorized as variations in exonic regions, splicing sites, intronic regions, upstream and downstream regions, and intergenic regions. Those in exons were further classified into synonymous or non-synonymous SNPs.

**Identification of Indels, CNVs, and SVs**. Similar to SNP calling, the calling of INDELs was conducted using SAMtools with minimum depth ≥4 and GQ >20, and only INDELs <100 bp were retained. CNVs were detected using both CNVnator v.0.3.2 (ref. [53]) and DELLY v.0.7.9 (ref. [54]). For CNVnator, the analyses were performed on the BAM files with a bin size of 100 bp and with the length >200 bp (Supplementary Methods). For DELLY, the analyses were conducted with default parameters, and deletions and duplications were considered to be CNVs (Supplementary Methods). Only the CNV calls with >50% of their lengths being overlapped between the two approaches were retained in the final set of CNVs. The CNVs that overlapped with gaps or genomic repeats were removed, and the remaining CNVs were segregated into short CNV bins (≥100 bp) across the genomes among the 248 individuals for subsequent analyses[25].

SVs were identified through the Manta v.1.6.0 (ref. [55]) and DELLY v.0.7.9 (ref. [54]). The two software called SVs by performing mapped paired-end reads and split reads analyses, and were run with default parameters to detect deletions (DEL), inversions (INV), duplications (DUP), and translocations (TRA) (Supplementary Methods). The SURVIVOR v.1.0.6 (ref. [56]) was implemented to detect the overlapping SVs identified by the two approaches with the command line ./SURVIVOR merge sample_files 1000 2 1 1 0 50 sample_merge.vcf.

**Population genetics analysis**. After filtering, we generated a set of SNPs for the following analyses. First, an individual-based neighbor-joining (NJ) tree was constructed for all the samples based on the nucleotide *p*-distance matrix using the TreeBeST v.1.9.2 (ref. [57]). The NJ tree was rooted with the outgroup of 16 Asiatic mouflon and visualized using the FigTree v.1.4.3 (http://tree.bio.ed.ac.uk/software/figtree/). PCA of whole-genome SNPs for all 248 individuals was performed with the GCTA v.1.24.2 (ref. [58]). Furthermore, population structure was assessed using the default setting in the ADMIXTURE v.1.23 (ref. [59]). The number of assumed genetic clusters *K* ranged from 2 to 7. To construct a NJ tree for the four subgroups of domestic sheep (i.e., African, East Asian, Middle Eastern, and European groups; see "Results"), 1–2 individuals from each landrace and two individuals from each sampling site of Asiatic mouflon were selected, totaling 50 individuals (i.e., 10 for each group; Supplementary Table 17). SNPs for the 50 individuals were extracted from the dataset of landraces and Asiatic mouflon. To mitigate the possible effect of LD, we implemented LD pruning using the parameter –indep-pairwise (50 5 0.4) in PLINK v.1.07 (ref. [60]). To eliminate the potential influence of selective SNPs, we only retained the SNPs located 150 kb away from genes and without missing genotypes. Eventually, a final set of 59,943 SNPs for the 50 individuals were kept for the construction of NJ tree. Reynolds genetic distances (ref. [61]) among the five

groups were calculated using Arlequin v3.5.2.2 (ref. [62]) (Supplementary Table 18). A NJ tree was then constructed based on the Reynolds genetic distances with 1000 bootstraps using PHYLIP v.3.695 (ref. [63]) and visualized using FigTree v.1.4.3. The parameter $r^2$ (ref. [64]) for LD was calculated for pairwise SNPs within each chromosome using PLINK v.1.07[60] with the parameters (–ld-window-$r^2$ 0 –ld-window 99999 –ld-window-kb 500). The average $r^2$ values were calculated for each length of distance and the whole-genome LD was averaged across all chromosomes. The LD decay plot was depicted against the length of distance using the *R* script (http://www.r-project.org). Nucleotide diversity ($\pi$) and global $F_{ST}$ were calculated using the vcftools v.0.1.14 (ref. [65]). The $F_{ST}$ values between populations were estimated using the ARLSUMSTAT implemented in the Arlequin v3.5.2.2 (ref. [62]), with a sliding window of 50 kb. The genomic SNP data of variant call format (VCF) were converted into the Arlequin format (arp) using the VCF2Arlequin python script[62]. Statistical significance (*P* values) of the $F_{ST}$ values were tested through 100,000 Markov chains following 10,000 burn-in steps. The average $F_{ST}$ and associated *P* values over all sliding windows were regarded as the values at the whole-genome level.

**Estimates of effective population size**. We used the PSMC[66] method to estimate changes in effective population size ($N_e$) over the last one million years. The PSMC analysis was implemented in each of the 248 samples. The parameters were set as follows: -N30 -t15 -r5 -p '4 + 25*2 + 4 + 6', with the filtering criteria of read depth for each SNP as six at the individual level. An average mutation rate ($\mu$) of $2.5 \times 10^{-8}$ per base per generation and a generation time (*g*) of 3 years[67] were used for the analysis. We also inferred recent $N_e$ using the SNeP v.1.0 (ref. [68]) with default settings. SNPs with missing data and a MAF smaller than 5% were excluded from the analysis. The different SNP marker distance bins for $r^2$ analysis were used to obtain different estimates of $N_e$ at $t = 1/2c$ generations ago.

**Detection of selective sweeps**. For SNPs, we performed tests for selective sweeps during domestication and breeding using two approaches based on the SNPs with less than 10% missing data: the XP-CLR approach implemented in the XP-CLR v.1.0 (ref. [69]) and by the comparison of $\pi$ ratios calculated using the vcftools v.0.1.14 (ref. [65]). To detect genomic regions under selection during domestication, we calculated the $\ln(\pi_{\text{-O. orientalis}}/\pi_{\text{-Landrace}})/\ln(2)$. Also, we estimated the ln $(\pi_{\text{-Control}}/\pi_{\text{-Target}})/\ln(2)$ between populations of domestic sheep with contrasting phenotypes for a specific target trait. The specific populations involved in comparisons between wild and domestic sheep and pairwise comparisons between domestic populations for detecting the signals associated with particular traits are shown in Supplementary Table 12. Values of $\pi$ were calculated with a 50 kb sliding window and a 25 kb sliding step. For the XP-CLR approach, a 0.5 cM sliding window with a spacing of 2 kb across the whole genomes were used for scanning, and 200 SNPs were assayed in each window with the parameters -w1 0.005 200 2,000 chrN -p0 0.95. To assess the statistical significance of the XP-CLR value for each window, we first estimated the proportion of SNPs with extreme XP-CLR values (i.e., top 1%) in the sliding windows, and then calculated the *P* values from the empirical distribution of the proportion scores obtained with these windows. In each comparison, the genomic regions in the top 1% XP-CLR values and ln($\pi$ ratio)/ln(2) values across the whole-genome were considered to be the selective sweeps.

Moreover, we estimated the *iHS* across the genomes of Asiatic mouflon and different groups of domestic sheep populations using the Selscan v.1.2.0 (ref. [70]) after filtering all missing data, with 50 kb sliding windows and 25 kb stepwise, a recombination rate of 1 cM Mb$^{-1}$ (ref. [9]) and default parameters –max-extend 1,000,000 –max-gap 200,000 –cutoff 0.05 (Supplementary Methods). We computed the proportions of SNPs with normalized |*iHS*| >2 in non-overlapping windows, and identified those windows within the top 5% empirical cutoff (i.e., above the 95th percentile of genome-wide distribution)[71] in the tested group as the signals of positive selection. We also employed the HKA test[72] to identify the selective signals associated with domestication and specific traits using Asiatic mouflon as an outgroup after filtering all missing data (Supplementary Methods). We calculated the $\chi^2$ statistic in 50 kb sliding windows and shift of 25 kb across the genome to find potential selective signals deviating from genome-wide neutral expectations. Two loci were analyzed each time, one was the 50 kb window taken from the tested genome and the other was the virtual neutral 50 kb window in terms of the average value of nucleotide statistics in the whole genome. After application of the HKA test for each sliding window, the $\chi^2$ statistic used to measure the goodness-of-fit was obtained and subsequently used to identify the selective signals.

For CNVs, we calculated a statistic $V_{ST}$, an analog to $F_{ST}$. $V_{ST}$ estimates population differentiation based on the quantitative intensity data and varies from 0 to 1 (ref. [25]). The statistic $V_{ST}$ of each CNV region was calculated to detect the selective signals between different comparisons[25]. $V_{ST}$ is defined as $(V_T - V_S)/V_T$, where $V_T$ is the variance of all the CNVs among all unrelated individuals in the target and control populations while $V_S$ represents the average variance in the target and control populations weighted for population size. The CNVs with the top 1% $V_{ST}$ values were considered as the selective CNVs.

**GWAS**. Association analyses of litter size, numbers of horns and nipples were performed using the MLM in the GEMMA v.0.96 (ref. [73]) based on a panel of 109,

146, and 123 samples collected from 11, 15, and 13 breeds, respectively (Supplementary Fig. 21). The effect of population stratification was corrected by adjusting the first three principal components (PCs) as derived from the whole-genome SNPs, and the proportion of variance explained by the markers was calculated using TASSEL v.5.0 (ref. [74]). To avoid potential false positives in multiple comparisons, the whole-genome significance threshold was adjusted via the Bonferroni test[75]. For SNPs, we set the thresholds as $-\log_{10}(P\ value) = 6, 6$, and 4 for litter size, numbers of horns and nipples, respectively. For CNVs, we set the thresholds as $-\log_{10}(0.05/total\ CNVs) = 5.28, 5.29$ for litter size and number of horns, respectively, but $-\log_{10}(P\ value) = 4$ for number of nipples using GEMMA v.0.96 based on the genotypes of CNVs selected by the DELLY and CNVnator. In addition, the quantile–quantile (Q-Q) plots of the MLM for individual traits were implemented in R Bioconductor.

**Permutation test for QTL overlaps.** We performed permutation test to check if the overlaps between selective sweeps/GWAS hits and QTL regions were significantly different from those expected at random. To this end, we used BEDTools v2.26.0 shuffle to generate simulated data sets by randomly selecting genomic regions of equal number and size to the observed selective sweeps/GWAS hits in the sheep genome, and we replicated this process 10,000 times[76]. We compared the number of overlaps between the observed selective sweeps/GWAS hits and the QTL regions with the distribution of overlap statistics between the simulated selective sweeps/GWAS hits data sets and the QTL regions, and calculated the statistical significance of $P$ values (i.e., the probability that a higher number of overlaps would be observed by chance).

**RNA-Seq analysis.** We collected tail adipose tissues from thin-tailed, short fat-tailed, long fat-tailed, and fat-rumped sheep for RNA-Seq analysis (Supplementary Table 13). Each tail type included three independent samples from different individuals as biological replicates. We used the Trizol RNA Reagent (Takara, Dalian, China) to extract total RNA from the tissues and measured the concentration and integrity of the RNA with the Agilent 2100 RNA 6000 Nano Kit (Agilent Technologies, Waldbronn, Germany). Subsequently, the libraries of mRNAs were constructed using the NEBNext® Ultra™ RNA Library Prep Kit (New England Biolabs, Ipswich, MA, USA) and sequenced using the Illumina HiSeq X Ten System to generate 150-bp paired-end reads. We used the HISAT v2.1.0, StringTie v2.0, and Ballgown package in R version 3.5.3 (ref. [77]) to map the paired-end reads to the sheep reference genome, assemble the reads, and estimate the gene expression levels, respectively. The number of reads matched to an expressed gene was standardized as fragments per kilobase of exon per million mapped fragments (FPKM) values. We employed the stattest function in the Ballgown package[77] to search for transcripts that were differentially expressed between the thin-tailed breed (Chinese Merino sheep) and fat-tailed/fat-rumped breeds (Small-tailed Han sheep, Large-tailed Han sheep, and Altay sheep), following correction for any differences in expression owing to population variables. This allowed us to get the confounder-adjusted fold changes between the two tested groups. The genes that exhibited $|\log_2(fold\ change)| \geq 2$ and adjusted $P \leq 0.05$ in the comparisons between fat-tailed/fat-rumped and thin-tailed individuals were considered as differentially expressed genes.

**Gene expression and western blot analyses of *PDGFD* gene.** We examined the gene expression level of *PDGFD* in the adipose tissues from the thin-tailed, short fat-tailed, long fat-tailed, and fat-rumped sheep through reverse transcription PCR (RT-PCR) and qPCR. The three biological replicates of adipose samples from different individuals for each tail type were used. The total RNA was extracted using the Trizol RNA Reagent (Takara, Dalian, China) and were treated with RNase-free DNase I to remove DNA using the RapidOut DNA Removal Kit (Thermo Fisher Scientific, Waltham, MA, USA). The first-strand cDNA was synthesized using the RevertAid First Strand cDNA Synthesis Kit (Thermo Fisher Scientific, Waltham, MA, USA). Following the manufacturer's instruction, 500 ng of RNA was reverse transcribed as the template for RT-PCR in 40 μl volume (including 20 μl RNA, 2 μl Random Hexamer Primer (100 μM), 8 μl 5× Reaction Buffer (including 250 mM Tris-HCl (pH 8.3), 250 mM KCl, 20 mM MgCl$_2$, 50 mM DTT), 2 μl RiboLock RNase Inhibitor (20 U μl$^{-1}$), 4 μl 10 mM dNTP Mix, 2 μl RevertAid RT (200 U μl$^{-1}$) and 2 μl nuclease-free water) and a thermocycling condition at 25 °C for 5 min, 42 °C for 60 min, and 70 °C for 5 min. Subsequently, the qPCR with SYBR Green (Promega, Madison, WI, USA) was performed on the QuantStudio™ 6 Flex Real-Time PCR System (Life Technologies, Carlsbad, CA, USA) using the first-strand cDNA and the primers designed based on the 5′- and -3′ end sequences of *PDGFD* gene (*PDGFD* F: 5′-GCGGATGCTCTGGACAAA and *PDGFD* R: AAGGAGGCAGCGTGGAAA-3′). The qPCR reactions and conditions were set as those described above. Each qPCR was run three times for one sample as technical replicates. Based on the qPCR results, the expression level was calculated using the $2^{-\Delta\Delta Ct}$ method[78] and normalized according to the internal control $\beta$-actin gene. The primers for $\beta$-actin were $\beta$-actin F (5′-CCAACCGT-GAGAAGATGACC) and $\beta$-actin R (CCCGAGGCGTACAGGGACAG-3′).

Protein was extracted from the tail adipose tissue using the Total Protein Extraction Kit (Huaxingbio, Beijing, China). The protein extract was mixed with an equal amount of sample buffer and then separated on 10% sodium dodecyl sulfate-polyacrylamide gel electrophoresis (SDS-PAGE) gels (60 μg per lane). The SDS-PAGE-separated proteins were electrophoretically transferred to a polyvinylidene fluoride (PDVF) membrane and then incubated for 3 hours at room temperature in blocking buffer (5% BSA in PBS-Tween 20). Immunodetection was carried out with the Rabbit Anti-beta Actin antibody (ab8227, Abcam, dilution 1:1,000), Anti-SCDGFB/PDGFD antibody (ab181845, Abcam, dilution 1:1,000) and Goat Anti-Rabbit IgG H&L (ab205718, Abcam, dilution 1:10,000). The blot signals were imaged using Tanon 6100 Chemiluminescent Imaging System and quantified using ImageJ (NIH) software. Photoshop CS6 were used to crop images from unprocessed images.

**Phenotyping.** Individual phenotype for traits, such as coat color, classes of fiber fineness, ear size, numbers of nipples and horns, and tail configurations were recorded for all the breeds whenever possible during sampling. Five different coat colors or color patterns, including white, white body with black head, black, brown, and gray were recorded for the animals sampled. The number of nipples ranged from 2 (normal) to 5 (selected). The horn phenotypes varied from polled to horned animals with 2–5 horns. The wool was graded into three classes (coarse, fine, and super fine) according to the British Wool Grading System (http://www.eytest.com/ey31f1.html). The tails were categorized into short fat-tailed, long fat-tailed, thin-tailed, and fat-rumped types according to the shape of tails of the animals as well as the recorded information for the breeds. Reproductive traits included number of litter per year, litter size in each birth, and seasonal or non-seasonal estrus extracted from the breeding records.

**Ethics statement.** All animal work was conducted according to a permit (no. IOZ13015) approved by the Committee for Animal Experiments of the Institute of Zoology, Chinese Academy of Sciences (CAS), China. For domestic sheep, animal sampling was also approved by local authorities where the samples were taken. For Asiatic mouflon, we collected peripheral blood samples from 14 captive Asiatic mouflon after receiving authorization for research from the Department of Environmental Protection in Iran (no. 93/34089). For other two Asiatic mouflon samples from Shahr-e Kord, Iran, sampling procedure was also approved by the governorate of Chaharmahal and Bakhtiari of Iran (no. 97.32.43.33165).

**Reporting summary.** Further information on research design is available in the Nature Research Reporting Summary linked to this article.

## Data availability

Raw sequencing data that support the findings of this study have been deposited to the NCBI BioProject database under accession PRJNA624020. The source data underlying Figs. 1, 3, 4, and 5 and Supplementary Figs. 1–7, 9, 11–21, and 24 are provided as a Source Data file.

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

## Acknowledgements
This study was financially supported by grants from the National Natural Science Foundation of China (nos. 31825024, 31661143014, and 31972527), the Special Funds of the State Key Laboratory of Sheep Genetic Improvement and Healthy Production (no. 2017CA001, 2018CA001, and 2019CA009), the External Cooperation Program of Chinese Academy of Sciences (152111KYSB20150010), the Second Tibetan Plateau Scientific Expedition and Research Program (STEP) (no. 2019QZKK0501), the National Key Research and Development Program-Key Projects of International Innovation Cooperation between Governments (2017YFE0117900), National Transgenic Major Program (2016ZX08008001), and the Taishan Scholars Program of Shandong Province (no. ts201511085). We express our thanks to the owners of the sheep for donating samples (see Supplementary Data 1) and Xue Ren for her help in sampling. For the Ethiopian populations, the sampling and DNA extraction were supported by the CGIAR Research Program on Livestock (CRP Livestock). Thanks are also owing to Amadou Traore (Institut de l'Environnement et de Recherches Agricoles (INERA), Ouagadougou, Burkina Faso), Masroor Ellahi Babar (Virtual University, Lahore, Pakistan), and Seyed Abbas Rafat (University of Tabriz, College of Agriculture, Tabriz, Iran) for their help during sample collection. The Chinese government contribution to CAAS-ILRI Joint Laboratory on Livestock and Forage Genetic Resources in Beijing is appreciated.

## Author contributions
M.-H. L. designed the study. M.-H.L., M.S., X.L.X., J.Y., F.W., E.H., Z.Q.S., Y.L.R., A.E., J.A.L., J.K., F.H.L., H.Y., Y.L.Y., C.B.L., P.Z., P.C.W., Y.S.Z., L.G., J.Q.Y., W.H.P., X.H.W., S.S.X., O.S., C.W., G.E., H.S.-D., M.D.Q., A.A., J.M.M., J.-L.H., K.P., and O.H. prepared the samples. X.L. and G.J.L. performed the genome data analyses. Y.X.X. and J.D. participated in the laboratory work. M.-H.L., X.L., J.Y., and X.L.X. wrote the manuscript with contributions from J.-L.H., J.A.L., M.W.B., D.W.C., and J.W.K.

## Competing interests
The authors declare no competing interests.

## Additional information

Xin Li[1,2,28], Ji Yang[1,3,28], Min Shen[4,5,28], Xing-Long Xie[1,2,28], Guang-Jian Liu[6,28], Ya-Xi Xu[1,3], Feng-Hua Lv[1,3], Hua Yang[4,5], Yong-Lin Yang[4,5], Chang-Bin Liu[4,5], Ping Zhou[4,5], Peng-Cheng Wan[4,5], Yun-Sheng Zhang[4,5], Lei Gao[4,5], Jing-Quan Yang[4,5], Wen-Hui Pi[4,5], Yan-Ling Ren[7], Zhi-Qiang Shen[7], Feng Wang[8], Juan Deng[1,9], Song-Song Xu[1,2], Hosein Salehian-Dehkordi[1,2], Eer Hehua[10], Ali Esmailizadeh[11], Mostafa Dehghani-Qanatqestani[11], Ondřej Štěpánek[12], Christina Weimann[13], Georg Erhardt[13], Agraw Amane[14,15], Joram M. Mwacharo[16], Jian-Lin Han[17,18], Olivier Hanotte[15,19,20], Johannes A. Lenstra[21], Juha Kantanen[22], David W. Coltman[23], James W. Kijas[24], Michael W. Bruford[25,26], Kathiravan Periasamy[27], Xin-Hua Wang[4,5✉] & Meng-Hua Li[1,3✉]

[1]CAS Key Laboratory of Animal Ecology and Conservation Biology, Institute of Zoology, Chinese Academy of Sciences (CAS), Beijing 100101, China. [2]University of Chinese Academy of Sciences (UCAS), Beijing 100049, China. [3]College of Animal Science and Technology, China Agricultural University, Beijing 100193, China. [4]Institute of Animal Husbandry and Veterinary Medicine, Xinjiang Academy of Agricultural and Reclamation Sciences, Shihezi 832000, China. [5]State Key Laboratory of Sheep Genetic Improvement and Healthy Breeding, Xinjiang Academy of Agricultural and Reclamation Sciences, Shihezi 832000, China. [6]Novogene Bioinformatics Institute, Beijing 100083, China. [7]Shandong Binzhou Academy of Animal Science and Veterinary Medicine, Binzhou 256600, China. [8]Institute of Sheep and Goat Science, Nanjing Agricultural University, Nanjing 210095, China. [9]College of Animal Science and Technology, Sichuan Agricultural University, Chengdu 611130, China. [10]Grass-Feeding Livestock Engineering Technology Research Center, Ningxia Academy of Agriculture and Forestry Sciences, Yinchuan, China. [11]Department of Animal Science, Faculty of Agriculture, Shahid Bahonar University of Kerman, Kerman, Iran. [12]Institute of Molecular Genetics of the ASCR, v. v. i., Vídeňská 1083, 142 20, Prague 4, Czech Republic. [13]Institute of Animal Breeding and Genetics, Justus Liebig University, Giessen, Germany. [14]Department of Microbial, Cellular and Molecular Biology, Addis Ababa University, Addis Ababa, Ethiopia. [15]LiveGene Program, International

Livestock Research Institute, Addis Ababa, Ethiopia. [16]Small Ruminant Genomics, International Centre for Agricultural Research in the Dry Areas (ICARDA), Addis Ababa, Ethiopia. [17]CAAS-ILRI Joint Laboratory on Livestock and Forage Genetic Resources, Institute of Animal Science, Chinese Academy of Agricultural Sciences (CAAS), Beijing, China. [18]Livestock Genetics Program, International Livestock Research Institute (ILRI), Nairobi, Kenya. [19]School of Life Sciences, University of Nottingham, University Park, Nottingham NG7 2RD, UK. [20]Center for Tropical Livestock Genetics and Health (CTLGH), the Roslin Institute, University of Edinburgh, Easter Bush, Midlothian EH25 9RG Scotland, UK. [21]Faculty of Veterinary Medicine, Utrecht University, Utrecht, the Netherlands. [22]Production Systems, Natural Resources Institute Finland (Luke), FI-31600 Jokioinen, Finland. [23]Department of Biological Sciences, University of Alberta, Edmonton, Alberta T6G 2E9, Canada. [24]CSIRO Livestock Industries, St Lucia, Brisbane, QLD, Australia. [25]School of Biosciences, Cardiff University, Cathays Park, Cardiff CF10 3AX Wales, UK. [26]Sustainable Places Research Institute, Cardiff University, CF10 3BA Cardiff, Wales, UK. [27]Animal Production and Health Laboratory, Joint FAO/IAEA Division of Nuclear Techniques in Food and Agriculture, International Atomic Energy Agency, Vienna, Austria. [28]These authors contributed equally: Xin Li, Ji Yang, Min Shen, Xing-Long Xie, Guang-Jian Liu. ✉email: wangxinhua5751@163.com; menghua.li@cau.edu.cn

