## [Peer Review File · Nature Communications]

Editorial Note: This manuscript has been previously reviewed at another journal that is not operating a transparent peer review scheme. This document only contains reviewer comments and rebuttal letters for versions considered at Nature Communications .

Reviewers' comments:

Reviewer #1 (Remarks to the Author):

This is a comprehensive genome work on sheep domestication and breeding. The amount of results is overwhelming but it is precisely because of that that the work is very difficult to read, it lacks focus, contains far too much supplementary data. Therefore, it requires complete rewriting with a few clearly defined messages in mind. A number of methods are applied one after the other but the final purpose is not clear.

I suggest to focus on a few strongly justified analyses. For instance, I am not sure merging SNPs with CNVs and SVs is needed. Different estimates of say variability are given, yet it seems appropriate estimates should be corrected for missing data so why to provide two estimates? How were some potential selective sweeps identified, eg, cold resistance? With so many phenotypes studied but also relatively few individuals for every phenotype, the amount of false positives is likely to be very high.

Experimental validation of snps and even of cnvs is ok but contributes to lengthening the ms without really being essential since there will always be false positives and negatives. SNP validation is done with genotyping arrays. FDR is decreased by combining both methods, but How many true snps do you lose by using only snps called by samtools and gatk?

I200 whether number of snps is or not significant does not make sense, since depend on number of animals as well as on variability.

I273: how do you correct for missing data? the estimate must consider % missing data

I2929: how were these 'p-values' obtained? They do not seem p-values

I333: HKA test, if Asiatic mouflon is the same species as *Ovis aries*, they cannot be strictly considered as outgroup, a better outgroup would be goat.

Reviewer #2 (Remarks to the Author):

General comments

Li et al. present the results of whole-genome resequencing of 249 wild and domestic sheep. The authors used plenty of breed-specific phenotypes to identify genes associated with domestication and artificial selection. The investigated phenotypes are only partly recorded at individual level. The quantity and quality of the here investigated material, applied methods and achieved results are of a high level and deserve a publication in a high impact journal like Nature Communications or Nature Genetics. The authors considered almost all my comments from a previous submission and improved the manuscript considerably.

However, the main issue still concerns the focus of the study. As the reader, I got the feeling of being crushed by the amount of results and after reading about half of the results, I got the feeling that I am in a loop. The authors applied comparable methods for two kinds of markers and used these to perform tests by contrasting different groups of breeds. They present the results for each method, each kind of marker and each population comparison one after another. I read this manuscript from the beginning to the end only and only because I am the reviewer. As a usual reader, I would stop reading in the middle of the long "Results" section (585 lines) and I would

switch to "Discussion". However, I would then notice that the discussion is very short (45 lines) without a real synthesis of the entire results. This is because the authors mostly combined the presentation of their own results with the discussion of these. At least for me, this was confusing and at many places in the manuscript, I was not sure what is a factual result of this study and

what is interpretation?

I suggest some improvements in the presentation of the results. There are different possibilities and I would suggest to follow the well-established path of separating the results and their discussion.

Please put in the "Results" section a factual presentation of the results supported by your own data. Please reduce opinions about the achieved results to a minimum in this section.

Please use the "Discussion" for a synthesis of the entire results and for an interpretation of these in the literature/expectations context.

Please check the possibility to order the results by tested hypothesis rather than by used method or kind of marker, e.g. all methods applied to different markers and breed comparisons but used to test the same hypothesis could be presented in parallel in the same section without a loop.

Sporadically I got the feeling that the authors want to impress the readers with the quantity of investigated phenotypes. I am impressed, however, in my opinion the quantity is part of the problem. In my opinion, there are too many results for a well-structured and focused study. Some breed comparisons or hypothesis testing can be omitted without loss of quality. There are many local breeds with short descriptions in books and on the Internet. These breed descriptions are provided by focal points or by breeders and their associations to book editors or to website administrators. According to our own experiences, this information relies on subjective opinions and does not rely on any systematic recording. For example, many owners and breeders of old, robust local breeds are pretty sure that their animals are highly disease resistant. However, this is a subjective opinion and not an objective and scientifically useful information. To map selection signatures associated with disease resistance, the authors compare the SOL breed against SFK, DPS, SHE and DRS. I was not able to find objective observations that were used to classify the SOL breed as disease-resistant and other breeds as low-resistant. In my opinion, comparative analyses of SOL against [SFK, DPS, SHE and DRS] increase only the quantity of the results but reduce the quality of this study.

There are other breed comparisons that lack objective classification. I was not able to find objective observations that were used to classify the quality of sheepskin, cold resistance, stress resistance and behavioural characteristics. To increase the focus on the most important achievements of this study, the authors should completely exclude the unused, unjustified or unnecessary phenotypes from the manuscript.

Specific comments

Line 218-219: Please use generations to provide more generally applicable information.

Line 241-254: This could be discussed in the sense of results that suggest domestication as a quantitative trait affected mostly by non-coding regions (structural and other mutations) rather than by nonsynonymous substitutions (<https://www.ncbi.nlm.nih.gov/pubmed/25170157/>, <https://www.ncbi.nlm.nih.gov/pubmed/28135247/>).

Line 273-281: Higher nucleotide diversity is observed in European sheep compared to Asian sheep; is this pattern expected? I would expect the opposite scenario because the bottleneck effect due to breed formation would be more pronounced in European breeds compared to the Asian breeds.

Line 278-280: I cannot follow and do not share this conclusion. Argali ($2n=56$) and Asiatic mouflon ($2n=54$) have different numbers of chromosomes and these two subspecies are geographically separated by Urial with $2n=58$. Which information did you use to conclude that the same chromosome chunks introgressed from Asiatic mouflon into Argali (at an unknown time in evolution) were forwarded to Bashibai sheep (at another unknown time in evolution)?

According to our own experiences, TreeMix is a very good tool to present a tree without migration,

but the results of TreeMix with some migration cannot be automatically taken as a fact. By our experiences, as soon as you change some parameter in your design (change starting random number or take one population out or include one new population into the design), the migration event estimated by TreeMix can change to another position or disappear.

Line 285-286: I cannot follow and do not share your conclusion about a separate domestication of fat-tailed sheep. The literature you cited also does not support this conclusion. Analyses of complete mtDNA genomes will not support such a conclusion. Your sequence data included mtDNA genomes, too. If you extract mtDNA reads of all sequenced mouflons (not only from this study but also additional publically available Asiatic mouflons), you will see that the diversity of mtDNA lineages covers all fat-tailed and thin-tailed sheep and you will find some lines not present in today's domestic sheep worldwide. This is not a suggestion to expand your study. This is a suggestion only to be careful with conclusions that are not supported by your or by published data.

Line 324-329: The demographic models as selected (model-I) based on ABC analysis contradict the theory of migration of Neolithic farmers along with their livestock in mainland Europe. Possibly you should discuss this.

Line 330-331: The above-mentioned dispersal of the fat-tailed sheep is not a fact.

Line 340: Seventeen Asiatic mouflon? Figure 1b suggested that you sequenced one Asiatic mouflon as a duplicate, i.e. the same sample was sequenced twice. In some other figures you correctly omitted the duplicated one (Supplementary Figure 3: 16 Asiatic mouflon). Did you exclude the duplicated mouflon from all analyses? If not, then this must be done. Please correct Material and methods appropriately, i.e. 16 Asiatic mouflon are investigated. This could be a major comment, too.

Line 341-344: I checked only two sources of information presented in these lines. I checked reference 3 for the description of the Drenthe Heath Sheep, which was "formed ~6,000 years ago" according to your text. It is written in reference 3: "Drentse Heideschaap (Local names or syn.: Drenthe Heath Sheep), established in 1800, is found in eastern Netherlands, Province Drente. It is of local origin being a composite of Drente Heath Sheep and Schoonebeker. The animals are white with black, grey, brown or yellow spots with a reddish brown or black head. Adult males weigh on average 50 kg and females 45 kg with an average wither height of 52 cm and 47 cm respectively. These sheep have coarse/carpet type wool and are horned, although there may be an occurrence of rare polled animals. This breed is reported to have mostly one lamb and is the only horned heather sheep breed in Netherlands. There are 804 females registered in the herd book, of which 90% are bred to males of the same breed. The in situ conservation programme involves 8 reproducing males and 2 herds."

I cannot follow and do not believe in the correct use of information from reference 3!

Furthermore, you wrote, "African Djallonké (~7,000 years; <https://www.livestockoftheworld.com/Sheep/>)". I checked this reference, too, and read the following: "Djallonké are a hairy thin-tailed type of sheep that originated in western Asia, and entered Africa through the Isthmus of Suez and Bab el Mandeb. Until the third Millennium BC, the hairy thin-tailed sheep was the only type of sheep on the African continent. Domestic sheep had reached Egypt and other parts of North Africa by 5000 BC (Epstein 1971)."

It is still a puzzle to me how you concluded that the Djallonké sheep breed was established 7,000 years ago? I only read general information that domesticated sheep reached North Africa 7,000 years ago.

I am not able check the proper use of all references in your manuscript but according to the above two and according to the unorthodox use of references 17&18 in Lines 278-280 and references 19-21 in Lines 285-286, I am very alarmed. This could be a major comment, too.

Lines 404 ff.: I did not understand your permutation test. You generated 10,000 simulated data sets by randomly permuting the genomic locations of selective sweeps/GWAS hits on the sheep genome.

I am not sure if you asked the correct question?

Suppose you have 100 selection signatures and you ask if these significantly overlap with a specific set of QTLs. To answer this question, we use a random number generator to get 100 random positions across the genome. These random positions have the same properties (length etc.) as 100 detected selection signatures. Then, by using the same settings, you check if these 100 random genome fragments overlap with QTLs. You used identical QTL overlapping criteria for this test, too. You asked the same question 10,000 times and checked the probability to get overlapping with QTLs by chance. I am not sure if permutation can answer this question?

Line 507-508: Is this overlapping significant?

Line 596: Is this overlapping significant?

Line 877-880: Was any filtering in the SNP datasets (based on read depth) carried out before performing PSMC?

Reviewers' comments:

Reviewer #1 (Remarks to the Author):

This is a comprehensive genome work on sheep domestication and breeding. The amount of results is overwhelming but it is precisely because of that that the work is very difficult to read, it lacks focus, contains far too much supplementary data.

Therefore, it requires complete rewriting with a few clearly defined messages in mind. A number of methods are applied one after the other but the final purpose is not clear.

- Good comment! As suggested also by reviewer 2, we have deleted the relevant parts related to the analyses and results of the traits such as cold resistance, disease resistance and sheepskin in the main text and supplementary information (including supplementary figs. 20-22 and relevant contents in supplementary figure 23 and supplementary tables 47-55 and 59-63 in the previous version). Please see the changes (highlighted by red color) in the manuscript text file. Also, as indicated above in our response to the comment by the editor, we have relocated part of the results and discussion to supplementary information. The revised version has been shortened greatly, and become concise and clear.

- We have quite much supplementary data, while the data are useful for relevant studies. For example, we have presented the information of samples, SNP and CNV statistics, SNP and CNV validation, results of selective analyses and GWAS, overlaps between detected signatures and known QTLs, and ABC modeling. Also, we have presented relevant discussions on SNP calling based on different reference genomes (i.e., Asiatic mouflon and domestic sheep), LD, effective population size, PSMC, demographic history and genomic differentiation etc in supplementary information. We think the information is useful for similar future investigations.

- Based on a few clearly defined messages, we have reorganized the manuscript and completely rewritten the manuscript. We separated the results and the discussion, and reorganized the results by presenting a set of methods and their purpose in the same section, which makes the main text concise and clear.

- We have now focused on the genetic mechanisms underlying traits during domestication and genetic improvement such as tail configuration and three other quantitative traits. Briefly, we have deleted contents about the traits of cold resistance, disease resistance and sheepskin. The analyses based on CNVs (i.e., selective analysis and GWAS analysis), the relevant parts about the experimental validation of SNPs and CNVs, and the results of permutation test for QTLs overlaps have been moved to the supplementary information. Please see the relevant changes marked in the revised main text and supplementary information.

- Regarding the methods and their purpose, to explore the genetic structure, we performed NJ tree, PCA and Bayesian clustering (STRUCTURE) analyses, and also calculated LD, recent effective population size, PSMC, nucleotide diversity and pairwise F_{st} , etc. To explore the genetic mechanism underpinning the traits of tail type and horn number, we performed selective analysis and GWAS analysis to identify candidate functional genes and variants. Our results of the genes and variants associated with distinct traits under

selection during domestication and genetic improvement will be a useful resource for sheep and other domestic animals. In this sense, our methods and the final purpose are clear in the revised version.

I suggest to focus on a few strongly justified analyses. For instance, I am not sure merging SNPs with CNVs and SVs is needed. Different estimates of say variability are given, yet it seems appropriate estimates should be corrected for missing data so why to provide two estimates? How were some potential selective sweeps identified, eg, cold resistance? With so many phenotypes studied but also relatively few individuals for every phenotype, the amount of false positives is likely to be very high.

- Good comment again! We have focused on the analyses of tail type and three quantitative traits in the revised version.

- We implemented analyses of different variants including SNPs, CNVs and SVs to detect selective signals in different variant levels (e.g., nonsynonymous substitutions, synonymous substitutions and structural variations) but not to mutual confirmation. We consider that SNPs, CNVs and SVs are different kinds of genetic variants, which may reveal distinct genetic mechanisms for sheep domestication and phenotypic traits, therefore merging SNPs with CNVs and SVs is necessary. Nevertheless, to make the manuscript concise, we described mainly the results of SNPs and the most prominent findings about CNVs and SVs in the main text. We moved other relevant contents of CNVs and SVs to supplementary information. Please see lines 233-279, 322-330, 410-434 and 587-605 in the supplementary information.

- Following the comments of reviewer 1 in the last round of review, we provided two different estimates of genomic variability (i.e., π values), one estimate is based on SNPs with less than 10% missing rate (a common filter criterion applied in many genomic studies) and the other is based on SNPs with no missing site in any individual (suggested by the reviewer 1 in the last round of review). In this version, we modified the relevant sentence (please see line 177-180 in the revised manuscript) to make the meaning clear.

- For SNPs, to ensure the reliability of selective analysis, we implemented four methods (XP-CLR, π ratio, iHS and HKA). For XP-CLR and π ratio, we used SNPs with less than 10% missing rate to analyze. For iHS and HKA, we used SNPs with no missing site in each population. Please see lines 794-795, 815-816 and 824 in the revised manuscript.

- According to the comments also suggested by reviewer 2, we have removed the relevant parts about the traits of cold resistance, disease resistance and sheepskin from the manuscript. Moreover, for the trait of tail type focused in this study, we performed experimental validation on gene expression level and protein level for important candidate genes (e.g., PDGFD). For another focused trait "horn number", we conducted GWAS analysis (146 individuals involved) and found many common candidate genes with those identified by the selective sweeps. Additionally, we also examined genotype mode of the candidate genes for other traits in a large panel of populations. We expect that these mutually verified analyses would greatly reduce potential false positives in the trait-associated selective sweeps.

Experimental validation of snps and even of cnvs is ok but contributes to lengthening

the ms without really being essential since there will always be false positives and negatives. SNP validation is done with genotyping arrays. FDR is decreased by combining both methods, but How many true snps do you lose by using only snps called by samtools and gatk?

- In the last round of review, reviewer 2 suggested to perform experimental validation for SNPs and CNVs. We think this is a regular and important way to inspect the quality of genomic data. To reduce the length of the manuscript as suggested by you, we have moved the section of experimental validation to supplementary information.

- Taking the SNPs with concordant genotypes between sequenced data and Ovine Infinium HD SNP BeadChip data in 223 individuals as an example, we identified an average of 298,415 SNPs and 305,725 SNPs for Samtools and GATK, respectively, and we consider these SNPs are true SNPs. After combining the two methods, we got an average of 237,432 true SNPs. Then we can measure that $(298,415-237,432)/298,415 = 20.44\%$ true SNPs were lost for Samtools and $(305,725-237,432)/305,725 = 22.34\%$ true SNPs were lost for GATK. Please see Supplementary Table 8.

- Using one method (Samtools/GATK) can get more SNPs but the accuracy cannot be guaranteed, whereas using the common sites of two methods can significantly decreased FDR (from 5.356% (Samtools), 6.375% (GATK) to 1.709% (Samtools & GATK)) in our dataset. Actually, there is always a paradox for SNP calling, some researcher selected the SNP data called by one of the approaches (Yano et al. 2016) and some selected the SNPs called by both approaches (Zhou et al. 2015; Cao et al. 2016).

- Although we lost SNPs when we combined Samtools and GATK, we indeed got a final set of 28.36 million common SNPs, with an average of 274 SNPs in each sweep (50-kb). We consider the quantity of the SNPs is adequate for reliable analyses. Furthermore, we attempt to provide accurate information for each nonsynonymous substitution which located in candidate genes, so we chose to perform the SNP calling by combining the two approaches and get the common sites to ensure the accuracy of our data. Please also see the relevant references below.

*Yano, K. et al. Genome-wide association study using whole-genome sequencing rapidly identifies new genes influencing agronomic traits in rice. Nat Genet. **48**, 927-934 (2016).*

*Zhou, Z. et al. Resequencing 302 wild and cultivated accessions identifies genes related to domestication and improvement in soybean. Nat Biotechnol. **33**, 408-414 (2015).*

*Cao, K. et al. Genome-wide association study of 12 agronomic traits in peach. Nat Commun. **7**, 13246 (2016).*

l200 whether number of snps is or not significant does not make sense, since depend on number of animals as well as on variability.

- In the last round of review, reviewer 2 asked us to compare the abundance of genetic variants in different populations and groups and further to test the significance. Here, we delete the significant test in the comparison of shared SNPs in this version based on your comments. Please see lines 171-173 in the revised manuscript.

l273: how do you correct for missing data? the estimate must consider % missing data

- We filtered the SNPs with the parameter "--mis 0.1", which means the SNP with more than 10% missing rate in all individuals will be filtered. Please see line 692 in the revised manuscript. Then we used these filter SNPs with less than 10% missing rate to calculate nucleotide diversities (π). Please see lines 177-179 and 223-225 in the revised manuscript.

- Following the suggestion of reviewer 2 in the last round of review, we also computed nucleotide diversities (π) after filtering the SNPs with missing genotypes in any individual, please see lines 179-180 and 225-226 in the revised manuscript.

1929: how were these 'p-values' obtained? They do not seem p-values

- Yes, they are not regular p-values. In fact, they are empirical cutoffs for the top 5% of strongest selective signals genome-wide. Such empirical cutoffs (e.g., top 1%, 5%, etc.) were proposed by the original paper introducing the iHS method (Voight et al. 2006) and were also used in other studies (e.g., "empirical P" in Kothapalli et al. 2016; "percentile" in Crawford et al. 2017).

- To make our description accurate, we removed these empirical p-values throughout the manuscript and modified the sentence to "We computed the proportions of SNPs with normalized $|iHS| > 2$ in non-overlapping windows, and identified those windows within the top 5% empirical cutoff (i.e., above the 95th percentile of genome-wide distribution) in the tested group as the signals of positive selection". Please see lines 818-822 in the revised manuscript. Also, please see the relevant references below:

Voight, B.F., et al. A map of recent positive selection in the human genome. *PLoS Biol.* 4, e72 (2006).

Kothapalli, K.S., et al. Positive selection on a regulatory insertion-deletion polymorphism in *FADS2* influences apparent endogenous synthesis of arachidonic acid. *Mol. Biol. Evol.* 33, 1726–1739 (2016).

Crawford, N.G., et al. Loci associated with skin pigmentation identified in African populations. *Science* 358, eaan8433 (2017).

1933: HKA test, if Asiatic mouflon is the same species as *Ovis aries*, they cannot be strictly considered as outgroup, a better outgroup would be goat.

- Asiatic mouflon (*Ovis orientalis*) and domestic sheep (*Ovis aries*) are in the same genus, but they are different species. Also, phylogenetic tree and PCA analyses showed clear phylogenetic differentiation between Asiatic mouflon and domestic sheep (Fig. 1b, c), therefore we think Asiatic mouflon as an outgroup is reasonable.

Reviewer #2 (Remarks to the Author):

General comments

Li et al. present the results of whole-genome resequencing of 249 wild and domestic sheep. The authors used plenty of breed-specific phenotypes to identify genes associated with domestication and artificial selection. The investigated phenotypes are only partly recorded at individual level.

The quantity and quality of the here investigated material, applied methods and

achieved results are of a high level and deserve a publication in a high impact journal like Nature Communications or Nature Genetics. The authors considered almost all my comments from a previous submission and improved the manuscript considerably.

- Thank you for the positive comments!

However, the main issue still concerns the focus of the study. As the reader, I got the feeling of being crushed by the amount of results and after reading about half of the results, I got the feeling that I am in a loop. The authors applied comparable methods for two kinds of markers and used these to perform tests by contrasting different groups of breeds. They present the results for each method, each kind of marker and each population comparison one after another. I read this manuscript from the beginning to the end only and only because I am the reviewer. As a usual reader, I would stop reading in the middle of the long “Results” section (585 lines) and I would switch to “Discussion”. However, I would then notice that the discussion is very short (45 lines) without a real synthesis of the entire results. This is because the authors mostly combined the presentation of their own results with the discussion of these. At least for me, this was confusing and at many places in the manuscript, I was not sure what is a factual result of this study and what is interpretation?

- Following your comments, we reorganized the results and pinpointed those with strong justification, high quality and great scientific significance. We have separated the results and the discussion, and reordered the results by presenting a set of methods and their purpose in the same section. Specifically, we moved sample information in results to methods section (see lines 648-654 in the revised manuscript), the discussion on genomic variation, , ratio of nonsynonymous and synonymous substitutions, domestication, global F_{ST} for improvement and tail configuration in the results section into discussion section (lines 543-556, 559-581, 592-604 and 614-631 in the revised manuscript) and moved discussion on F_{ST} among different geographic populations, LD among wild and domestic sheep and divergence time of N_{ME} and N_{EA} into supplementary information (lines 332-347 and 358-382 in the revised supplementary information).

- Notably, we have moved all analysis based on CNVs/SVs and QTLs to supplementary information (lines 322-330, 410-434 and 587-650 in the revised supplementary information), which greatly shorten the length of main text.

- Now we clearly present in the text that the main focus of this study is to explore the genetic structure and demographic history of domestic sheep and the genetic mechanisms underlying traits during domestication and genetic improvement such as tail configuration and three other quantitative traits. Please see also responses above.

- As far as the findings of this study, firstly we found that the genetic structure of sheep is consistent with its geographical distribution, which is the same with previous studies (Kijas et al. 2009) and explored the demographic history of sheep. Then we identified 36 reliable domesticated genes which is overlapped with recent two studies and also found 22 new candidate genes associated with domestication in sheep. For the tail configuration, we focus on the PDGFD gene, and identified this gene plays an inhibitory roles in adipogenesis. For horn number, GWAS and selective analyses showed that RXFP2 gene is highly related

with polledness and we identified eight significant SNPs at the upstream of RXFP2 gene (reverse orientation in the reference genome), five significant SNPs and one significant CNV at the downstream of RXFP2 gene (Fig. 5 and Supplementary Table 62). Furthermore, the two SNPs with frequency distinction among breeds with different horn number (Fig. 5) also located in the downstream of RXFP2 rather than the coding region, which indicated that the polledness trait is affected mostly by non-coding regions (structural and other mutations) rather than by nonsynonymous substitutions in coding regions (Carneiro et al. 2014; Medugorac et al. 2017).

Kijas, J.W. et al. A genome wide survey of SNP variation reveals the genetic structure of sheep breeds. PLoS One. 4, e4668 (2009).

Carneiro, M. et al. Rabbit genome analysis reveals a polygenic basis for phenotypic change during domestication. Science 345, 1074–1079 (2014).

Medugorac, I. et al. Whole-genome analysis of introgressive hybridization and characterization of the bovine legacy of Mongolian yaks. Nat. Genet. 49, 470–475 (2017).

I suggest some improvements in the presentation of the results. There are different possibilities and I would suggest to follow the well-established path of separating the results and their discussion.

Please put in the “Results” section a factual presentation of the results supported by your own data. Please reduce opinions about the achieved results to a minimum in this section.

Please use the “Discussion” for a synthesis of the entire results and for an interpretation of these in the literature/expectations context.

Please check the possibility to order the results by tested hypothesis rather than by used method or kind of marker, e.g. all methods applied to different markers and breed comparisons but used to test the same hypothesis could be presented in parallel in the same section without a loop.

- Excellent comments! We have placed the results and the discussion in separated sections. Specifically, we only presented the results obtained from our own data in the “Results” section, and added interpretations of the main results to the “Discussion” section in the literature/expectations context. Please see our response above.

- As suggested, we have ordered the results by the tested hypothesis such as genomic signatures related to domestication, breeding and improvement, etc.

- Also, different kinds of markers and methods used for testing the same hypothesis have been presented in parallel in the same section. Please see relevant sections in the revised version.

Sporadically I got the feeling that the authors want to impress the readers with the quantity of investigated phenotypes. I am impressed, however, in my opinion the quantity is part of the problem. In my opinion, there are too many results for a well-structured and focused study.

Some breed comparisons or hypothesis testing can be omitted without loss of quality.

There are many local breeds with short descriptions in books and on the Internet. These

breed descriptions are provided by focal points or by breeders and their associations to book editors or to website administrators. According to our own experiences, this information relies on subjective opinions and does not rely on any systematic recording. For example, many owners and breeders of old, robust local breeds are pretty sure that their animals are highly disease resistant. However, this is a subjective opinion and not an objective and scientifically useful information. To map selection signatures associated with disease resistance, the authors compare the SOL breed against SFK, DPS, SHE and DRS. I was not able to find objective observations that were used to classify the SOL breed as disease-resistant and other breeds as low-resistant. In my opinion, comparative analyses of SOL against [SFK, DPS, SHE and DRS] increase only the quantity of the results but reduce the quality of this study.

There are other breed comparisons that lack objective classification. I was not able to find objective observations that were used to classify the quality of sheepskin, cold resistance, stress resistance and behavioural characteristics. To increase the focus on the most important achievements of this study, the authors should completely exclude the unused, unjustified or unnecessary phenotypes from the manuscript.

- As suggested, we have removed the relevant parts about disease resistance, cold resistance, the quality of sheepskin, stress resistance and behavioural characteristics from the main text and supplementary information in the updated version. Please see the tracks of the changes in the revised manuscript text.

Specific comments

Line 218-219: Please use generations to provide more generally applicable information.

- We have changed the sentence into "It should be noted that the nearly identical nucleotide diversity between landraces and improved breeds ... has only been in practice over the around 200 generations (Kijas et al. 2012)", Please see lines 546-549 in the revised manuscript.

Kijas, J.W. et al. Genome-wide analysis of the world's sheep breeds reveals high levels of historic mixture and strong recent selection. PLoS Biol. 10, e1001258 (2012).

Line 241-254: This could be discussed in the sense of results that suggest domestication as a quantitative trait affected mostly by non-coding regions (structural and other mutations) rather than by nonsynonymous substitutions

(<https://www.ncbi.nlm.nih.gov/pubmed/25170157/>,
<https://www.ncbi.nlm.nih.gov/pubmed/28135247/>).

- Good comment! We agree with you that domestication could have been affected by non-coding regions. Following the methods used in the reference you recommend (<https://www.ncbi.nlm.nih.gov/pubmed/25170157/>), we screened the domestication-associated sweeps detected in this paper to examine the quantity and location (i.e., coding or non-coding region) of fixed (or nearly fixed) SNPs for derived alleles. Specifically, through combining the XP-CLR and π ratio analyses, we identified 209 domesticated selective regions and further obtained 260 fixed SNPs (derived allele frequency ≥ 0.95) in these regions. After annotation of the 260 fixed SNPs, we detected 79

SNPs located in mRNA region and only 7 SNPs located in exon region. Accordingly, we added some discussion in the sense of the results that “This suggested domestication as a quantitative trait (e.g., horn number) was affected mostly by mutations in non-coding regions rather than by nonsynonymous substitutions (Carneiro et al. 2014; Medugorac et al. 2017)”. Please see lines 566-570 in the revised manuscript.

Carneiro, M., et al. Rabbit genome analysis reveals a polygenic basis for phenotypic change during domestication. Science 345, 1074–1079 (2014).

Medugorac, I., et al. Whole-genome analysis of introgressive hybridization and characterization of the bovine legacy of Mongolian yaks. Nat. Genet. 49, 470–475 (2017).

Line 273-281: Higher nucleotide diversity is observed in European sheep compared to Asian sheep; is this pattern expected? I would expect the opposite scenario because the bottleneck effect due to breed formation would be more pronounced in European breeds compared to the Asian breeds.

- We agree with you that higher nucleotide diversity should be observed in Asian sheep. Our results showed opposite scenario was due to that we have calculated the estimate for the group of Asian populations and the group of European populations, rather than for each Asian and European sheep population. Here, we calculated nucleotide diversity of each population, and found that an average of nucleotide diversities (π) in Asian sheep is 2.817×10^{-4} , which is significantly ($P = 0.00000567$) higher than European sheep (2.485×10^{-4}). We added this information in the main text, please see lines 222-226 in the revised manuscript.

Line 278-280: I cannot follow and do not share this conclusion. Argali ($2n=56$) and Asiatic mouflon ($2n=54$) have different numbers of chromosomes and these two subspecies are geographically separated by Urial with $2n=58$. Which information did you use to conclude that the same chromosome chunks introgressed from Asiatic mouflon into Argali (at an unknown time in evolution) were forwarded to Bashibai sheep (at another unknown time in evolution)?

- Good comments. This sentence has been deleted.

According to our own experiences, TreeMix is a very good tool to present a tree without migration, but the results of TreeMix with some migration cannot be automatically taken as a fact. By our experiences, as soon as you change some parameter in your design (change starting random number or take one population out or include one new population into the design), the migration event estimated by TreeMix can change to another position or disappear.

- We agree with you that the results of TreeMix are changeable. In the last round of review, the reviewer 2 advised us to perform Treemix to test the hypothesis of gene flow, which showed similar results with Admixture. However, for the reason that we have no sufficient evidence to explain this introgressed event (from Asiatic mouflon to Bashibai sheep), we deleted the relevant results and discussion.

Line 285-286: I cannot follow and do not share your conclusion about a separate domestication of fat-tailed sheep. The literature you cited also does not support this conclusion. Analyses of complete mtDNA genomes will not support such a conclusion. Your sequence data included mtDNA genomes, too. If you extract mtDNA reads of all sequenced mouflons (not only from this study but also additional publically available Asiatic mouflons), you will see that the diversity of mtDNA lineages covers all fat-tailed and thin-tailed sheep and you will find some lines not present in today's domestic sheep worldwide. This is not a suggestion to expand your study. This is a suggestion only to be careful with conclusions that are not supported by your or by published data.

- We corrected the sentence : Since the dispersal of fat-tailed sheep occurred as a separate event after the first introduction of domestic sheep¹⁹⁻²¹. Please see lines 365-366 in the revised supplementary information.

Line 324-329: The demographic models as selected (model-I) based on ABC analysis contradict the theory of migration of Neolithic farmers along with their livestock in mainland Europe. Possibly you should discuss this.

- Since the postdomestic spread of the wool sheep have erased the pattern created by the first migrations, the Neolithic introduction of agriculture occurred earlier than indicated by our model. Also for European sheep, the time estimate (T_{EU}) is much more recent than the first introduction of agriculture and is probably influenced by the subsequent spreading of wool sheep replacing the original hair-sheep domesticates (Chessa et al. 2009). We have added this discussion in the main text, please see lines 378-382 in the revised supplementary information.

Chessa, B. et al. Revealing the history of sheep domestication using retrovirus integrations. Science. 324, 532-6 (2009).

Line 330-331: The above-mentioned dispersal of the fat-tailed sheep is not a fact.

- We have deleted it in the main text.

Line 340: Seventeen Asiatic mouflon? Figure 1b suggested that you sequenced one Asiatic mouflon as a duplicate, i.e. the same sample was sequenced twice. In some other figures you correctly omitted the duplicated one (Supplementary Figure 3: 16 Asiatic mouflon). Did you exclude the duplicated mouflon from all analyses? If not, then this must be done. Please correct Material and methods appropriately, i.e. 16 Asiatic mouflon are investigated. This could be a major comment, too.

- We compared the whole genome SNP genotype of 17 wild sheep and find the concordance rate between individuals numbered KR.15 and numbered 273 is 99.51% whereas between any other two individuals are around 56%. Therefore, we deleted the wild sheep numbered KR.15 with lower sequencing depth compared with the wild sheep number 273 and re-performed all related analysis. Please see the revised manuscript and supplementary information.

Line 341-344: I checked only two sources of information presented in these lines. I

checked reference 3 for the description of the Drenthe Heath Sheep, which was “formed ~6,000 years ago” according to your text. It is written in reference 3: “Drentse Heideschaap (Local names or syn.: Drenthe Heath Sheep), established in 1800, is found in eastern Netherlands, Province Drenthe. It is of local origin being a composite of Drenthe Heath Sheep and Schoonebeker. The animals are white with black, grey, brown or yellow spots with a reddish brown or black head. Adult males weigh on average 50 kg and females 45 kg with an average wither height of 52 cm and 47 cm respectively. These sheep have coarse/carpet type wool and are horned, although there may be an occurrence of rare polled animals. This breed is reported to have mostly one lamb and is the only horned heather sheep breed in Netherlands. There are 804 females registered in the herd book, of which 90% are bred to males of the same breed. The in situ conservation programme involves 8 reproducing males and 2 herds.”

I cannot follow and do not believe in the correct use of information from reference 3!
- *We agree with you that reference 3 was not properly used here. In the updated version, we cited a new reference (Ryder 1981, reference 12) which stated that the Drenthe Heath Sheep is a primitive European sheep considered to be of Iron Age type, and accordingly, we revised “formed ~6,000 years ago” to “appeared in the Iron Age (approximately, 800 BC – 50 BC in Europe)”. Please see lines 1017-1018 in the revised manuscript.*

Ryder, M. A survey of European primitive breeds of sheep. Ann. Genet. Sel. Anim. 13, 381–418 (1981).

Furthermore, you wrote, “African Djallonké (~7,000 years; <https://www.livestockoftheworld.com/Sheep/>”). I checked this reference, too, and read the following: “Djallonké are a hairy thin-tailed type of sheep that originated in western Asia, and entered Africa through the Isthmus of Suez and Bab el Mandeb. Until the third Millennium BC, the hairy thin-tailed sheep was the only type of sheep on the African continent. Domestic sheep had reached Egypt and other parts of North Africa by 5000 BC (Epstein 1971).”

It is still a puzzle to me how you concluded that the Djallonké sheep breed was established 7,000 years ago? I only read general information that domesticated sheep reached North Africa 7,000 years ago.

- *Following your comments, we searched for more convincing references for the history of the Djallonké sheep breed. In the paper of Yaro et al. (2016), it is written in page 44: “Certain African livestock breeds such as Djallonké sheep and Taurine cattle, which entered Africa from the near east around 5000 BCE and 7000 BCE respectively,”. This indicates that Djallonké sheep was established in Africa ~7,000 years ago. Moreover, in the paper of Muigai & Hanotte (2013), it was written in page 40: “They (i.e., the African long-legged sheep and the tropical dwarf sheep (Djallonké sheep) mentioned in previous sentence) are considered to be the most ancient type of sheep on the continent”. This also indicates that Djallonké sheep is an old African breed. Additionally, Epstein (1971) presented that Djallonké are hairy thin-tailed sheep and the hairy thin-tailed sheep is the*

only type of sheep on the African continent until the third Millennium BC. In the updated version, we cited these new references (references 14-16) and revised “~7,000 years” to “> 5,000 years”. Please see lines 1021-1027 in the revised manuscript.

Yaro, M., Munyard, K.A., Stear, M.J. & Groth, D.M. Combatting African Animal Trypanosomiasis (AAT) in livestock: The potential role of trypanotolerance. Vet. Parasitol. 225, 43–52 (2016).

Muigai, A.W.T. & Hanotte, O. The origin of African sheep: archaeological and genetic perspectives. Afr. Archaeol. Rev. 30, 39–50 (2013).

Epstein, H. The origin of the domestic animals of Africa. New York: Africana Publishing Corporation (1971).

I am not able check the proper use of all references in your manuscript but according to the above two and according to the unorthodox use of references 17&18 in Lines 278-280 and references 19-21 in Lines 285-286, I am very alarmed. This could be a major comment, too.

- For references 17 & 18 in lines 278-280, we have removed this discussion because it is not supported by available references. The references 19-21 actually provided the information about the dispersal of fat-tailed sheep, so we modified the sentence to “Since the dispersal of fat-tailed sheep occurred as a separate event after the first introduction of domestic sheep¹⁹⁻²¹”. Please see lines 365-366 in the revised supplementary information.

- We have went throughout the main text and supplementary materials and checked again all the references to see if they indeed support what the paper claims. We have corrected the improper use of references. For instance, we correct the reference 87 (the previous reference 2) “Ryder, M.L. Evolution of Domesticated Animals (Longman, New York, 1984)” to “Ryder, M.L. Sheep. In: Mason, LL., ed., Evolution of Domesticated Animals. Longman, New York, 63- 84 (1984)”. We cite a new reference (the reference 17) “Porter, V., Alderson, L., Hall, S.J.G. & Sponenberg, D.P. Mason’s Wold Encyclopedia of Livestock Breeds and Breeding: 2 volume pack (CAB International, Wallingford, 2016)” instead of the previous website to support that Karakul sheep is an old landrace. We removed one of the two duplicated references (the previous reference 36 and 82, and the reference 31 in this version). In addition, we also correct several small mistakes in the references 37, 80, 83 and 85. Please see lines 1077-1080, 1184-1185, 1191-1192 and 1196-1198 in the revised manuscript.

Lines 404 ff.: I did not understand your permutation test. You generated 10,000 simulated data sets by randomly permuting the genomic locations of selective sweeps/ GWAS hits on the sheep genome.

I am not sure if you asked the correct question?

Suppose you have 100 selection signatures and you ask if these significantly overlap with a specific set of QTLs. To answer this question, we use a random number generator to get 100 random positions across the genome. These random positions have the same properties (length etc.) as 100 detected selection signatures. Then, by using the same

settings, you check if these 100 random genome fragments overlap with QTLs. You used identical QTL overlapping criteria for this test, too. You asked the same question 10,000 times and checked the probability to get overlapping with QTLs by chance. I am not sure if permutation can answer this question?

- *As you commented, the detailed process in our permutation test is as follows: we used BEDTools “shuffle” (Quinlan & Hall 2010; Jain et al. 2018) to generate simulated data sets by randomly selecting genomic regions of equal number and size to the observed selective sweeps/GWAS hits in the sheep genome, and we replicated this process 10,000 times. Then, we compared the number of overlaps between the observed selective sweeps/GWAS hits and the QTL regions with the distribution of overlap statistics between the simulated selective sweeps/GWAS hits datasets and the QTL regions, and calculated the statistical significance of P-values (i.e., the probability that higher number of overlaps would be observed by chance).*

- *Take reproductive traits as an example, we detected a total of 29 selective sweeps by XP-CLR, π ratio and iHS methods and found 5 sweeps are overlapped with 3 reproductive QTLs, then we ask if these 5 sweeps significantly overlap with the 3 QTLs. To answer this question, we randomly selected 29 fragments with the same settings including length and non-overlapping as the observed 29 selective sweeps from the whole genome using the software Bedtools with shuffle command (Quinlan & Hall 2010; Jain et al. 2018). Then we check if the number of overlapping fragments between 29 random genome fragments and the 3 reproductive QTLs is more than 5. Repeatedly, we asked the same question 10,000 times and then checked the probability to get higher number (> 5) of overlaps with QTLs by chance. Finally, we obtained a p-value < 0.01 .*

- *We have carefully considered if our permutation tests as described in lines 859-868 in the revised manuscript is appropriate for answering our question. As explained more clearly in the revised version (please see our response to this comment above), we indeed carried out the permutation tests as described by the reviewer. We believe that this gives the chance of overlap if sweep/GWAS regions with QTLs if their respective distributions are independent. In this sense, we think the permutation implemented can answer the question. In fact, similar permutation tests have been also used in other studies (Cheng et al. 2016; Price et al. 2018; Wu et al. 2018).*

*Quinlan, A.R. & Hall, I.M. BEDTools: a flexible suite of utilities for comparing genomic features. *Bioinformatics* **26**, 841–842 (2010).*

*Cheng, F., et al. Subgenome parallel selection is associated with morphotype diversification and convergent crop domestication in *Brassica rapa* and *Brassica oleracea*. *Nat. Genet.* **48**, 1218–1224 (2016).*

*Jain, M., et al. Nanopore sequencing and assembly of a human genome with ultra-long reads. *Nat. Biotechnol.* **36**, 338–345 (2018).*

*Price, N., et al. Combining population genomics and fitness QTLs to identify the genetics of local adaptation in *Arabidopsis thaliana*. *Proc. Natl. Acad. Sci. USA* **115**, 5028–5033 (2018).*

*Wu, J., et al. Diversification and independent domestication of Asian and European pears. *Genome Biol.* **19**, 77 (2018).*

Line 507-508: Is this overlapping significant?

- Not significant, specific P value ($P = 0.5684$) is shown in supplementary table 25.

Line 596: Is this overlapping significant?

- Overlappings are significant ($P < 0.05$) in all traits except for wool traits ($P = 0.2254$). To check the specific P values and relevant information, please see supplementary table 25 and lines 613-630 in the revised supplementary information.

Line 877-880: Was any filtering in the SNP datasets (based on read depth) carried out before performing PSMC?

- Before performing PSMC, we generate input file in fastq format with the commands “samtools mpileup -q 1 -C 50 -S -D -m 2 -F 0.002” and “vcfutils.pl vcf2fq -d 6 -D 90 -Q 20”. Therefore, the criteria of read depth for each SNP is 6 at the individual level. We have added this filtering information in the method section, please see line 770 in the revised manuscript.

Reviewers' comments:

Reviewer #1 (Remarks to the Author):

The ms has improved and could be potentially publishable, although I am uncomfortable with some analyses; these can be safely removed without affecting the whole story. Some other text is also probably unnecessary.

- Variability: I mentioned this before but is uncorrected. Tajima's Theta is number of average differences between sequences divided by length sequenced. In order to account for missing data, you need specific algorithms (eg, ferreti et al Genetics). In general though, counting number of snps is a good proxy with similar depths across samples as is here. I suggest then to report only one measure of variability, and not to repeat the results (lines 179 and 223). Note that a variability 1×10^{-3} is human-like whereas 3×10^{-4} (no miss) is much much lower and clearly unrealistic.

- ABC: I am sorry to say the analyses reported is probably flawed. You report N_e with HPD ranges between 10k and 400k so basically a flat distribution. You do not report HPD for split times but should be in the same range. You do not consider migration either. Besides N_e in the thousands are valid may be for *Drosophila* at best but not for large mammals. I know by experience by experience that ABC fine tuning is devilishly complicated and unstable. Then, please remove all ABC methods and results, you can substitute for a simple NJ tree.

- Overall, outlier regions are taken as selective sweeps. Be more cautious in describing them.

Line 89: NGS in animals has been applied as in plants for selective detection.

l97: genes potentially affected by selection.

l109ff: Not so many digits are needed 136.99  137 and so on. Billion is 10^{*9} I guess, change to scientific notation.

- l126-138: Is alignment against mufllon used for something? If not, delete all this.

- Voluntary: consider removing all GO related analyses and results.

- Age of breeds (l265): In agreement with second reviewer, I consider all these ages as unscientific and unproven. Breeds have interchanged animals for years.

- l277 putative selected regions.

-l451 potentially selected genes

- l480 suggesting a role <-- suporting the prominent role

- l551: I know of no case in evolution (may HIV ?) where $K_a > K_s$, least in humans, I am also highly concerned about how literature is cited as is referee 2!

-l568-570: This is simply not true and does not follow from the previous lines.

-l639-640 This looks like wishful thinking. Consider removing as I do not see how these kind of works can help small producers in developing countries.

IMPORTANT: SRA accessions must be included before publication. Uploading such a huge dataset can be a painful and slow process.

Reviewer #2 (Remarks to the Author):

General comments

As already mentioned in my last review, the quantity and quality of the here investigated material, applied methods and achieved results are of a high level and deserve a publication in a high impact journal like Nature Communications.

The authors considered almost all my comments. However, for my impression, the presentation of the story is still a little boring but this could be a matter of taste.

The reader will find some results and discussion only in the supplementary material. As reader, I check supplementary material only at very specific interest. Therefore, these results and in particular discussion are almost not present for me.

Specific comments

The artificial selection and isolation (formation of breeds) could have stronger effect on LD and N_e than nucleotide diversity. Authors presented LD and N_e results but do not discuss these in any direction.

Authors follow my suggestion and perform additional analyses to prove domestication as quantitative trait. However, authors use wrong example (horn number) to emphasise their results.

Reviewers' comments:

Reviewer #1 (Remarks to the Author):

The ms has improved and could be potentially publishable, although I am uncomfortable with some analyses; these can be safely removed without affecting the whole story. Some other text is also probably unnecessary.

- Variability: I mentioned this before but is uncorrected. Tajima's Theta is number of average differences between sequences divided by length sequenced. In order to account for missing data, you need specific algorithms (eg, ferreti et al Genetics). In general though, counting number of snps is a good proxy with similar depths across samples as is here. I suggest then to report only one measure of variability, and not to repeat the results (lines 179 and 223). Note that a variability 1×10^{-3} is human-like whereas 3×10^{-4} (no miss) is much much lower and clearly unrealistic.

- *We have deleted the repetitive results on nucleotide diversities (π) (lines 179-180 and 225-226 in the previous main text). In a previous study (Naval-Sanchez et al. 2018), variability for domestic sheep is 1.6×10^{-3} and for wild sheep is 2.0×10^{-3} based on the SNPs with less than 10% missing, which is similar with our results. when simply filtering all missing data, the nucleotide diversities (π) declined to an unrealistic level (e.g., 3×10^{-4}) because a lot of SNPs were lost. Therefore, we kept the results of the variability based on the SNPs with less than 10% missing and removed the results after filtering all missing data.*

Naval-Sanchez, M. et al. Sheep genome functional annotation reveals proximal regulatory elements contributed to the evolution of modern breeds. Nat. Commun. 9, 859 (2018).

- ABC: I am sorry to say the analyses reported is probably flawed. You report N_e with HPD ranges between 10k and 400k so basically a flat distribution. You do not report HPD for split times but should be in the same range. You do not consider migration either. Besides N_e in the thousands are valid may be for Drosophila at best but not for large mammals. I know by experience by experience that ABC fine tuning is devilishly complicated and unstable. Then, please remove all ABC methods and results, you can substitute for a simple NJ tree.

- *In the methodological algorithm of ABC, it is difficult to consider all the parameters in the demographic models, and the best-supported model may not exactly reflect the real scenario in the history. In this study, our ABC modeling analysis mainly focused on the divergence pattern (i.e., split times) of different genetic groups of domestic sheep, and did not consider more complicated scenarios, e.g., the influence of migration on divergence.*

- *We used proper methods (e.g., ABCtoolbox pipeline) to perform the ABC modeling analysis, and we did report the HPD for split times in the main text (lines 252-259),*

Supplementary Fig. 29 and Supplementary Table 17 in the previous version.

- Following the suggestion, we have removed all relevant ABC methods and results and added a NJ tree instead (Fig. 1e). The NJ tree was constructed using the same samples as in previous ABC analysis, and the branch length of each sheep group and the bootstrap value of each divergence node were shown on the tree. Please see lines 205-208 in the main text and Supplementary Tables 66 and 67. We also added the individuals and analyses in the section of Methods. Please see lines 705-718.

- Overall, outlier regions are taken as selective sweeps. Be more cautious in describing them.

- We used four methods (XP-CLR, π ratio, iHS and HKA) to identify selective signals, and considered those detected by at least two methods as putative selective sweeps. We therefore are confident with the results of selective sweeps. Nevertheless, we agree with your comments and have replaced “selective sweeps” with “putative selective sweeps” or “potentially selected sweeps” where appropriate. Please see the revised version of the manuscript.

Line 89: NGS in animals has been applied as in plants for selective detection.

- We have corrected the sentence into ‘So far whole-genome resequencing has allowed the identification of genomic variants involved in domestication and genetic improvement for several domestic plants (e.g., rice, maize and soybean)⁴⁻⁶ and animals (e.g., swine, cattle and sheep)^{1,2,7,8}.’ Please see lines 80-83 in the revised manuscript.

l97: genes potentially affected by selection.

- Revised.

l109ff: Not so many digits are needed 136.99  137 and so on. Billion is 10**9 I guess, change to scientific notation.

- We have checked throughout the whole manuscript, and have exhibited less digits. Please see the revised version of manuscript. In addition, We have checked relevant references (Varshney et al. 2017; Yano et al. 2016), they always described big data (e.g., paired-end reads) using “billion”, so we kept “billion” rather than changed to scientific notation.

Varshney, R.K. et al. Whole-genome resequencing of 292 pigeonpea accessions identifies genomic regions associated with domestication and agronomic traits. Nat. Genet. 49, 1082–1088 (2017).

Yano K. et al. Genome-wide association study using whole-genome sequencing rapidly identifies new genes influencing agronomic traits in rice. Nat. Genet. 48, 927-34 (2016).

- l126-138: Is alignment against mouflon used for something? If not, delete all this.

- In the first round of review, reviewer 2 suggested us to align the data against

Asiatic mouflon reference genome and examine the difference in using different reference genome. Because the Asiatic mouflon genome is still incompletely assembled, we didn't use the aligning results for any downstream analysis. Therefore, we have deleted this part in the main text and supplementary information following your comments.

- Voluntary: consider removing all GO related analyses and results.
- *GO related analyses and results still are useful information in the manuscript and for future relevant researches, so we consider to keep the main findings of them and just delete some concrete names of GO terms and KEGG pathways in the main text. Please see lines 289-292 and 330-335 in the revised manuscript.*

- Age of breeds (l265): In agreement with second reviewer, I consider all these ages as unscientific and unproven. Breeds have interchanged animals for years.
- *Although the detailed ages for the five breeds might not be well proved from an archaeological view, the literature and book we cited at least provide information that the five breeds are old landrace sheep. To avoid potential confusion, we removed the specific ages for the breeds and just kept the references cited in the sentence. Please see lines 233-234 in the revised manuscript.*

- l277 putative selected regions.
- *Revised.*

-l451 potentially selected genes
- *Revised.*

- l480 suggesting a role <-- supporting the prominent role
- *Corrected.*

- l551: I know of no case in evolution (may HIV ?) where $K_a > K_s$, least in humans, I am also highly concerned about how literature is cited as is referee 2!
- *We have carefully checked the literatures cited in previous supplementary table 69 and confirmed that they are correct. Nevertheless, the nonsynonymous/synonymous ratio for human (1.4226), cattle (1.2124) and yak (1.4008) were actually calculated in this study by using the number of nonsynonymous and synonymous SNPs reported in the literature (please see the original data in the tables below). To avoid confusion, we deleted the previous supplementary table 69 and relevant parts in the main text. Please see lines 508-510 in the revised manuscript.*

Table S2. Variation detection from Tibetan exomes

SNP discovery for functional classes of sites

Genomic features		Known	Novel	Total
		# of SNPs	# of SNPs	# of SNPs
CDS	synonymous	14,439	12,312	26,751
	nonsynonymous	11,421	26,634	38,055
	nonsense	73	541	614
Intron		14,547	23,623	38,170
5'UTR		848	1,129	1,977
3'UTR		895	1,100	1,995
Intergenic		15	16	31

Supplementary Table 2. Number of SNPs and Indels in each annotation class

Variant class	SNP	Indel
intergenic variant	18,589,752	1,074,289
intron_variant	6,579,341	391,250
upstream_gene_variant	673,577	42,687
downstream_gene_variant	592,605	38,778
missense_variant	99,089	63
synonymous_variant	81,730	
3_prime_UTR_variant	61,009	4,152
frameshift_variant		1,046
inframe_insertion		251
inframe_deletion		719
splice_region_variant	18,439	922
5_prime_UTR_variant	10,989	675
stop_gained	4,155	4
splice_donor_variant	2,268	78
non_coding_exon_variant	1,789	51
splice_acceptor_variant	1,628	93
initiator_codon_variant	184	
stop_lost	118	
coding_sequence_variant	101	39
stop_retained_variant	57	
mature_miRNA_variant	52	1
nc_transcript_variant	12	8
Total	26,716,895	1,555,106

Table 2 The distribution of SNPs in the yak genome

Location	All	Wild-specific SNPs
Intergenic	6 510 697	1 293 045
Gene region	1 868 889	392 223
Intron	1 785 609	378 151
Exonic	83 280	14 072
Nonsynonymous	48 591	7872
Synonymous	34 689	6200
Stop gain	765	111
Stop loss	26	2
Total	8 379 586	1 685 268

-1568-570: This is simply not true and does not follow from the previous lines.

- *Focused on the 260 fixed SNPs (derived allele frequency ≥ 0.95) in the 209 putative selective sweeps during domestication, we annotated them using Ovis.4.0.gff file and found 79 SNPs located in the mRNA region (coding region) and 7 SNPs in the exon regions in the last round of review. To make the results more accurate, we used the SNP dataset which were annotated using the package ANNOVAR v.2013-06-21 and identified 200 SNPs in the intergenic regions, 3 in the upstream and 3 in the downstream, while the remaining 54 SNPs in the coding regions comprise 2 SNPs in the exon and 52 in the intron.*

-*In addition, we also focused on 600 significant SNPs selected by GWAS analysis for litter size trait, and found 323 SNPs in the intergenic regions, one in the downstream, while the remaining 276 SNPs in the coding regions comprise only one SNP in the exon and 275 in the intron.*

-*Therefore, to make the meaning of the sentence clear, we have modified the sentence and its previous lines to “Nevertheless, by annotation of the 260 fixed SNPs (derived allele frequency ≥ 0.95) in the 209 putative selective sweeps, we detected 206 SNPs located in the non-coding regions and only 2 SNPs in the exon regions including one nonsynonymous mutation. This suggested domestication as a quantitative trait (e.g., litter size) was affected mostly by mutations in non-coding regions^{50,51}, while less mutations were nonsynonymous substitutions.” Please see lines 521-526 in the revised manuscript.*

-1639-640 This looks like wishful thinking. Consider removing as I do not see how these kind of works can help small producers in developing countries.

- Deleted.

IMPORTANT: SRA accessions must be included before publication. Uploading such a huge dataset can be a painful and slow process.

- Yes, uploading such a huge dataset (~20.55 Tb) is indeed a slow process. Instead of uploading the dataset to NCBI and obtaining SRA accessions, we plan to upload the genetic data to the National Genomics Data Center (NGDC)

(<https://bigd.big.ac.cn/gsa/>), which is more convenient for us and allowed by a variety of well-known journals (e.g., Nature, Science, Cell and Nature Communications).

Nature: <https://www.nature.com/articles/nature23883>

Science: <https://science.sciencemag.org/content/364/6440/eaau6389>

Cell:

[https://www.cell.com/cell/fulltext/S0092-8674\(17\)30713-4?returnURL=https%3A%2F%2Flinkinghub.elsevier.com%2Fretrieve%2Fpii%2FS0092867417307134%3Fshowall%3Dtrue](https://www.cell.com/cell/fulltext/S0092-8674(17)30713-4?returnURL=https%3A%2F%2Flinkinghub.elsevier.com%2Fretrieve%2Fpii%2FS0092867417307134%3Fshowall%3Dtrue)

Nature Communications: <https://www.nature.com/articles/s41467-019-12174-w>

Reviewer #2 (Remarks to the Author):

General comments

As already mentioned in my last review, the quantity and quality of the here investigated material, applied methods and achieved results are of a high level and deserve a publication in a high impact journal like Nature Communications.

- Thank you for the positive comments!

The authors considered almost all my comments. However, for my impression, the presentation of the story is still a little boring but this could be a matter of taste.

- Following the comments of reviewer 2 in the last round of review, we have separated the results and discussion which were previously an integrated section. In fact, there are some papers presented the results and discussion separately, whereas the other papers still presented the results and discussion jointly (Varshney et al. 2017; Zhou et al. 2018; Naval-Sanchez et al. 2018). Just as the reviewer said, the presentation of the story could be a matter of taste.

*Varshney, R.K. et al. Whole-genome resequencing of 292 pigeonpea accessions identifies genomic regions associated with domestication and agronomic traits. Nat. Genet. **49**, 1082–1088 (2017).*

*Zhou, Z. et al. An intercross population study reveals genes associated with body size and plumage color in ducks. Nat. Commun. **9**, 2648 (2018).*

*Naval-Sanchez, M. et al. Sheep genome functional annotation reveals proximal regulatory elements contributed to the evolution of modern breeds. Nat. Commun. **9**, 859 (2018).*

The reader will find some results and discussion only in the supplementary

material. As reader, I check supplementary material only at very specific interest. Therefore, these results and in particular discussion are almost not present for me.

- This manuscript has much information, so some minor points peripheral to the main findings were shown as the supplementary information. We think the supplementary information including figures, tables and notes are useful for relevant studies, and the readers can check the supplementary materials as they need.

Specific comments

The artificial selection and isolation (formation of breeds) could have stronger effect on LD and Ne than nucleotide diversity. Authors presented LD and Ne results but do not discuss these in any direction.

- In the previous version, we put the relevant discussion in the supplementary notes. Here we add the discussion as suggested in the main text. Please see lines 226-228 in the revised manuscript.

Authors follow my suggestion and perform additional analyses to prove domestication as quantitative trait. However, author use wrong example (horn number) to emphasise their results.

- Good comment! We have instead provided another example, i.e., litter size, to illustrate the results. In fact, litter size is a quantitative trait which was selected during domestication, and we found that by annotation of the 600 SNPs in the selective sweeps associated with litter size, we detected 324 SNPs located in the non-coding regions (including 323 SNPs in the intergenic regions and 1 in the downstream) and only one SNP in the exon regions. This could well support the results that domestication as a quantitative trait was affected mostly by mutations in non-coding regions, while less mutations were nonsynonymous substitutions. Please see lines 521-526 in the revised manuscript.

REVIEWERS' COMMENTS:

Reviewer #1 (Remarks to the Author):

No further comments. I believe that having access to raw data will make the article much more influential and citable.

Reviewers' comments:

Reviewer #1 (Remarks to the Author):

No further comments. I believe that having access to raw data will make the article much more influential and citable.

- *We are depositing the raw data to the NCBI BioProject database under accession PRJNA624020 [<https://www.ncbi.nlm.nih.gov/bioproject/PRJNA624020>].*